# Circle-RoPE: Cone-like Decoupled Rotary Positional Embedding for Vision-Language Models

**Chengcheng Wang** [1*]  **Jianyuan Guo** [3*]  **Hongguang Li** [2*]  **Yuchuan Tian** [4]  **Ying Nie** [2]  **Chang Xu** [1†]  **Kai Han** [2†]

[1]University of Sydney.  [2]Huawei Noah's Ark Lab.  [3]City University of Hong Kong.  [4]State Key Lab of General AI, School of Intelligence Science and Technology, Peking University.

cwan0785@uni.sydney.edu.au  jianyguo@cityu.edu.hk  lihongguang0014@gmail.com
tianyc@stu.pku.edu.cn  ying.nie@huawei.com  c.xu@sydney.edu.au  hankai@pku.edu.cn

## Abstract

Rotary Position Embedding (RoPE) is widely adopted in large language models, but when applied to vision-language models (VLMs) it couples text and image position indices and can introduce spurious cross-modal relative-position bias. We propose *Per-Token Distance* (PTD) to quantify cross-modal positional disentanglement, and we prove that PTD $= 0$ is a sufficient condition to eliminate the geometric attention bias induced by RoPE. Guided by this criterion, we introduce Circle-RoPE, which remaps 2D image-token coordinates onto an annulus orthogonal to the text position axis, yielding a cone-like geometry where each text token is equidistant to all image tokens while preserving intra-image spatial structure. We further propose Alternating Geometry Encoding (AGE) to synergize complementary geometric priors by alternating the decoupled geometry of Circle-RoPE and the grid-based prior of standard RoPE across layers. This design ensures both rigorous cross-modal disentanglement and the preservation of fine-grained intra-image spatial structure, and experiments on diverse VLM backbones and multimodal benchmarks show consistent gains in spatial grounding and visual reasoning. The code is available at https://github.com/lose4578/CircleRoPE.

## 1. Introduction

Rotary Position Embedding (RoPE) (Su et al., 2024) has become a widely used choice for encoding relative positional information in large language models (LLMs). When ex-

tending such models to handle both textual and visual inputs, as in vision-language models (VLMs), a central challenge is how to represent positional information across heterogeneous modalities. Text is inherently sequential, whereas visual information is spatially structured and characterized by attributes such as location, orientation, viewpoint, and scale—properties that are fundamentally different from, and largely uncorrelated with, textual order.

Different approaches have been explored to tackle this issue. For instance, Figure 2(a) illustrates models such as LLaVA (Liu et al., 2023a), Emu3 (Wang et al., 2024b), InternLM-VL (Chen et al., 2024), and DeepSeek-VL2 (Wu et al., 2024), which flatten image tokens into a 1D sequence and concatenate them with text tokens, then directly apply the standard 1D RoPE from LLMs to the resulting multimodal token sequence. Figure 2(b) shows the strategy used in mPLUG-Owl3 (Ye et al., 2024), where all image patches are assigned the same image token index. Figure 2(c) shows M-RoPE (Wang et al., 2024a) (Qwen2-VL), which preserves the spatial layout of images while modeling textual sequentiality, but still concatenates image and text tokens in a single token sequence as in Figure 2(a).

Existing RoPE variants typically (i) flatten visual tokens into a 1D sequence or (ii) assign them 2D grid indices, and then concatenate them with text tokens. In both cases, the resulting cross-modal relative positions are largely determined by the positional-indexing scheme rather than semantic alignment, thereby **inducing spurious cross-modal positional bias** that can hinder multimodal understanding.

Figure 1 illustrates this issue with a visual question answering (VQA) example: "What type of religion is displayed high on the clock tower?" The phrase *"high on"* requires spatial grounding, and *"clock tower"* requires object recognition. However, their relative positions to the relevant image regions can be distorted by index-based encoding. Two common failure modes are: (i) semantic misalignment—*"high on"* should align with the top of the tower (index 1) but is instead closer (in index space) to an unrelated patch (index 8);

---

*Equal contribution. †Corresponding authors.

*Proceedings of the $43^{rd}$ International Conference on Machine Learning*, Seoul, South Korea. PMLR 306, 2026. Copyright 2026 by the author(s).

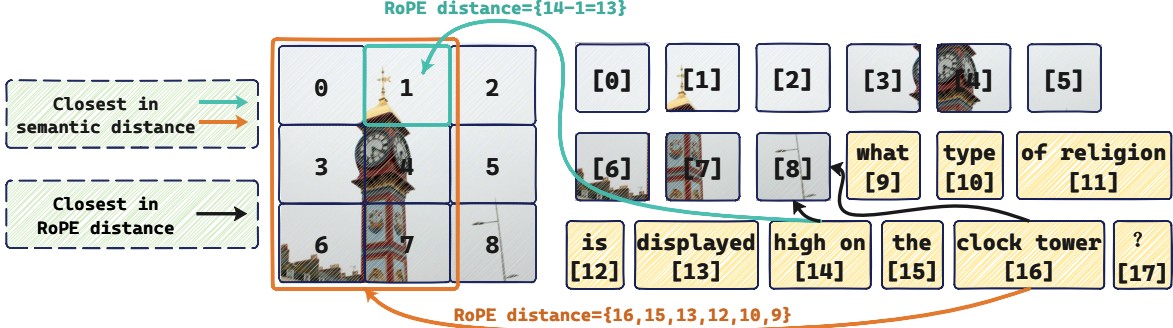

*Figure 1.* A VQA example where image and text tokens are sequentially concatenated. The image token at index 8 exhibits the smallest RoPE distance to all text tokens, despite semantically closer image tokens being located elsewhere. The text token at index 16 exhibits varying distances to the six image patches that correspond to the same semantic content. These misalignments highlight how conventional RoPE methods introduce spurious cross-modal relative positional bias.

and (ii) inconsistent multi-token distances—*"clock tower"* corresponds to multiple image tokens, yet their relative distances to the same text span vary, yielding inconsistent cross-modal cues.

In this work, we approach this problem from a geometric first principle: **how should a textual "observer" be positioned relative to a visual "canvas" in the embedding space?** We argue that the ideal geometric relationship is orthogonality. Imagine the text token as an observer and the image tokens forming a 2D plane. If the observer is placed coplanar with the image (as in 1D flattening), "perspective distortion" occurs—some image patches become artificially closer (smaller RoPE distance) than others simply due to their rasterization order, creating spurious positional cues.

To eliminate this bias, we propose **Circle Rotary Position Embedding (Circle-RoPE)**. Inspired by the geometry of a right circular cone, we map image tokens onto a base annulus and place the text token on the orthogonal normal axis (the cone's height). This cone-like geometry guarantees that each text token remains **equidistant** to all image tokens (satisfying our $\mathrm{PTD} = 0$ condition), thereby creating an isotropic attention field where attention is driven by semantic relevance rather than positional proximity.

Specifically, we extend M-RoPE, which represents image token indices using height–width coordinates, with two key innovations. First, we propose *Circular Image Token Index Projection* (**CIP**, Sec. 4.1), which maps 2D grid coordinates onto an annulus in 3D space whose normal direction is aligned with the text token index axis. This transformation enforces an orthogonal separation: each text token index lies on the normal axis and maintains equal Euclidean distance (and thus consistent RoPE distance) to all points on the annulus, forming a cone-like geometry. Meanwhile, relative spatial relationships among image tokens are preserved, as shown in Figure 2(d). This design disentangles positional dependencies across modalities.

Second, we propose *Alternating Geometry Encoding* (**AGE**, Sec. 4.2), which alternates between M-RoPE and Circle-RoPE across layers to leverage their complementary inductive biases and obtain more robust multimodal representations.

To make these claims precise, we summarize our contributions as follows:

- We identify and empirically illustrate *spurious cross-modal positional bias* induced by coupled indexing in existing RoPE-based multimodal encodings.

- We introduce the *Per-Token Distance* (PTD) metric to quantify cross-modal positional disentanglement, and we provide a formal guarantee (Appendix A) that $\mathrm{PTD} = 0$ is sufficient to eliminate the geometric attention bias in RoPE.

- We propose Circle-RoPE (CIP+AGE), which enforces consistent text–image relative positions while preserving intra-image spatial structure.

- We validate Circle-RoPE on multiple VLM backbones and diverse multimodal benchmarks, demonstrating improved spatial grounding and visual reasoning.

**Conflict of Interest Disclosure.** The authors declare no financial conflicts of interest related to this work. All models, datasets, and evaluation resources used in this paper are publicly available, and the authors have no financial sponsorship, employment-related evaluation conflict, or other financial relationship that could reasonably be perceived as influencing the results.

## 2. Related Work

Vision-language models (VLMs) unify visual and textual representations within a single transformer, and positional

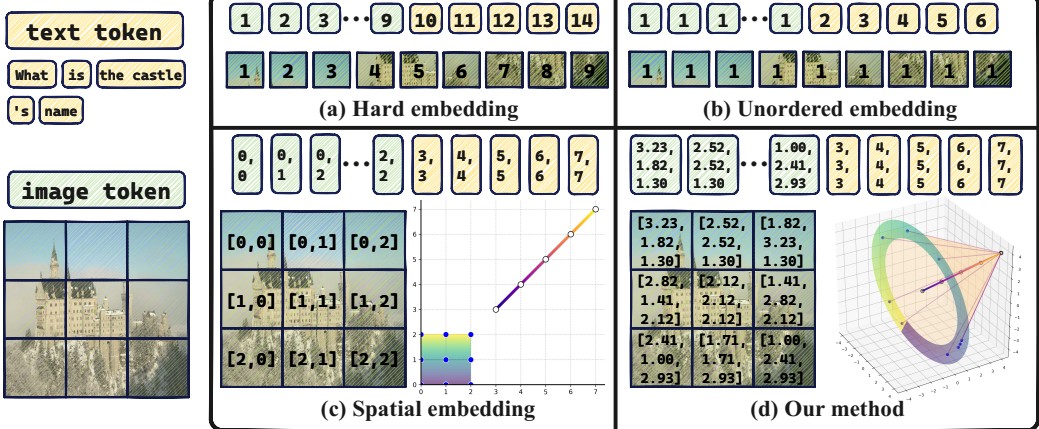

*Figure 2.* Text (yellow) and image (green) tokens are labeled with their position indices under different RoPE-based encoding schemes. (a) Hard embedding: image tokens are flattened into a 1D sequence and indexed accordingly. (b) Unordered embedding: all image tokens within an image share the same image token index. (c) Spatial embedding: image tokens are indexed by their 2D positions in the original image. (d) Ours: image token indices are remapped onto an annulus orthogonal to the text token index direction, yielding a decoupled encoding.

encoding is crucial for aligning heterogeneous modalities. Recent progress in multimodal LLMs and pixel-level understanding further emphasizes the importance of unified architectures and effective positional encoding strategies (Fei et al., 2025; Zhang et al., 2025; Lei et al., 2025; Yuan et al., 2025; Li et al., 2024; Wei et al., 2025). Broader model-design research has also explored architecture search for diffusion models, hierarchical long-context attention, hallucination mitigation in LVLMs, and adaptive reasoning for vision-language-action models (Li et al., 2023; Zeng et al., 2026; Li et al., 2025a; 2026b;a).

Many influential VLMs apply RoPE (Su et al., 2024) on a single concatenated token sequence. For example, LLaVA (Liu et al., 2023a;b), Emu3 (Wang et al., 2024b), InternLM-VL (Chen et al., 2024), Baichuan-Omni (Li et al., 2025b), Eve (Rang et al., 2025), and DeepSeek-VL2 (Wu et al., 2024) flatten image patches into a 1D sequence and concatenate them with text tokens before applying RoPE (or its variant) across modalities. Although widely adopted in practice, this design ties cross-modal relative positions to the chosen indexing and concatenation scheme.

To reduce cross-modal index disparity, mPLUG-Owl3 (Ye et al., 2024) assigns a shared position index to all visual tokens (via a placeholder token) when applying RoPE. This strategy partially alleviates modality-mixing bias, but it also removes fine-grained intra-image positional distinctions.

To better preserve image structure, Qwen2-VL introduces Multimodal RoPE (M-RoPE) (Wang et al., 2024a) by decomposing RoPE into separate dimensions for visual tokens (e.g., height, width, and temporal indices for video), while maintaining 1D indices for text. Although spatial RoPE variants preserve intra-image relationships more faithfully

than 1D flattening, cross-modal relative positions can still remain coupled through the shared index space.

Overall, existing RoPE-based VLMs either sacrifice intra-image spatial structure (shared-index) or retain cross-modal coupling (concatenation-based and spatial RoPE variants). In the next section, we revisit these designs from a unified mechanism perspective (Figure 2) and formalize their cross-modal bias via the proposed PTD metric.

## 3. Preliminaries and Problem Analysis

Although modern VLMs adopt different RoPE-based designs, a common root cause of cross-modal positional bias is that text token indices and image token indices are embedded into a shared index space. To make the discussion concrete, we categorize typical multimodal RoPE schemes according to how they assign indices to image tokens (Figure 2):

- **Hard embedding** (Figure 2(a)): Image tokens are first flattened into a 1D sequence and then concatenated with text tokens, after which standard 1D RoPE is applied on the joint sequence. However, because RoPE encodes relative positions through index offsets, the resulting text–image relative positions are largely determined by the concatenation order and the absolute index gap rather than semantic correspondence, which can introduce spurious cross-modal positional bias.

- **Unordered embedding** (Figure 2(b)): All image tokens within an image are assigned the same position index, so the RoPE-index distance from any text token to all image tokens becomes identical. This design can mitigate part

of the modality-mixing bias, but it also collapses all intra-image relative positions, discarding fine-grained spatial structure and harming spatial grounding.

- **Spatial embedding** (Figure 2(c)): Image tokens are assigned 2D indices based on their spatial positions in the image, which preserves intra-image relative structure and avoids extremely large 1D indices. Nevertheless, because image and text tokens are still embedded in a shared index space (via concatenation), cross-modal relative positions remain coupled and positional independence between modalities is not guaranteed.

In this section, we formalize the above issue and introduce a metric that quantifies the degree of cross-modal positional disentanglement.

*Table 1.* PTD values of different RoPE methods.

| Embedding method | Hard | Unordered | Spatial | Ours |
|---|---|---|---|---|
| Relative position information | ✔ | ✗ | ✔ | ✔ |
| PTD | 2.22 | 0 | 0.64 | **0** |

Existing approaches predominantly focus on encoding spatial information for images and sequential information for text independently, while overlooking the interference introduced by coupled cross-modal indexing. This coupling can distort cross-modal alignment by injecting spurious positional cues. Ideally, to eliminate such effects, the RoPE-index distance between each text token and the set of image tokens should be consistent, thereby enforcing positional independence across modalities.

**Per-Token Distance (PTD) Metric.** To quantify and compare how different RoPE-based methods affect cross-modal relative positions between text and image tokens, we introduce *Per-Token Distance* (PTD). Conceptually, for each text token $t$, we expect its distances (in the RoPE index space) to all image tokens to be as uniform as possible; otherwise, the model may receive spurious cross-modal positional signals unrelated to semantics. PTD measures the variance of the distances from each text token to the set of image tokens, and thus serves as a proxy for cross-modal positional disentanglement. Formally, let the index set of image tokens be $I = \{i_1, i_2, ..., i_{N_{\text{image}}}\}$ and the index set of text tokens be $T = \{t_1, t_2, ..., t_{N_{\text{text}}}\}$. PTD is defined as:

$$\bar{D}_t = \frac{1}{N_{\text{image}}} \sum_{i \in I} d(t, i) \tag{1}$$

$$\text{PTD} = \frac{1}{N_{\text{image}} N_{\text{text}}} \sum_{t \in T} \sum_{i \in I} \left| d(t, i) - \bar{D}_t \right| \tag{2}$$

where $d(x, y)$ denotes the Euclidean distance between indices $x$ and $y$ in the RoPE index space. A smaller PTD value indicates lower variance in the distances from each

text token to the set of image tokens, suggesting a higher degree of positional disentanglement across modalities.

**Connection to RoPE attention.** Let $m_t$ and $m_i$ denote the RoPE indices of a text query token $t$ and an image key token $i$. With RoPE, their attention logit can be written as

$$s_{t,i} = (\mathcal{R}_{m_t} \mathbf{q}_t)^\top (\mathcal{R}_{m_i} \mathbf{k}_i) = \mathbf{q}_t^\top \mathcal{R}_{\delta_{t,i}} \mathbf{k}_i,$$
$$\delta_{t,i} \triangleq m_i - m_t. \tag{3}$$

Under mild and explicit assumptions stated in Appendix A, the *semantic-conditioned expected logit* admits a distance-dependent alignment kernel:

$$\mathbb{E}[s_{t,i} \mid \text{sem}(t, i)] \approx A \cdot \mathcal{D}(r_{t,i}),$$
$$r_{t,i} \triangleq \|\delta_{t,i}\|_2 = \|m_i - m_t\|_2. \tag{4}$$

where $A > 0$ is a semantic scale and $\mathcal{D}(\cdot)$ is a (typically monotone) RoPE alignment kernel on the relevant range. Therefore, if $\text{PTD} > 0$, then for some $t$ the distances $\{r_{t,i}\}_{i \in I}$ are not all equal, which induces non-uniform expected logits purely due to RoPE index geometry, i.e., *geometric attention bias*. Conversely, if $\text{PTD} = 0$, then for every fixed $t$ all $r_{t,i}$ are identical across $i \in I$, so the RoPE-induced expected contribution is constant over image tokens and cannot encode any positional preference. Appendix A further shows that a logit-level bias measure is bounded (up to constants) by $\text{PTD}$.

We compute PTD for three typical multimodal encoding methods, *i.e.*, hard embedding (Figure 2(a)), unordered embedding (Figure 2(b)), and spatial embedding (Figure 2(c)). For convenience, we set $N_{\text{image}} = 9$ and $N_{\text{text}} = 5$. The PTD values are shown in Table 1.

Therefore, we propose to map all image token indices to positions that are equidistant to every text token index, aiming to minimize PTD (ideally to 0) and mitigate spurious cross-modal positional bias.

## 4. Method

We propose a novel positional encoding method for VLMs, Circle Rotary Position Embedding (Circle-RoPE). Circle-RoPE applies geometric transformations to image token indices $(w, h)$ prior to the RoPE rotation, with the goal of removing spurious cross-modal relative positional bias while preserving intra-image spatial relationships. Circle-RoPE consists of two components: *Circular Image Token Index Projection* (**CIP**, Sec. 4.1), which enforces cross-modal positional disentanglement (targeting $\text{PTD} = 0$), and *Alternating Geometry Encoding* (**AGE**, Sec. 4.2), which synergizes complementary geometric priors by alternating Circle-RoPE (for decoupled cross-modal geometry) and M-RoPE (for grid-based local spatial priors) across layers.

## 4.1. Circular Image Token Index Projection

Guided by the PTD criterion, we design *Circular Image Token Index Projection* (CIP) to decouple image token indices from text token indices, with the explicit goal of achieving $\mathrm{PTD} = 0$. CIP constructs a "cone-like" geometry for image indices so that, for each text token, the RoPE-index distance to the *set* of image tokens becomes constant, removing spurious cross-modal positional bias. At the same time, CIP avoids collapsing all image tokens to a single index: it preserves intra-image structure via an angle design on an annulus and uses a radius $R$ to control the overall spatial scale.

CIP consists of the following three steps:

(i) *Coordinate Centralization*: Shift the geometric center of all image token indices to the origin, standardizing the coordinate reference.

(ii) *Mixed-Angle Annular Mapping*: Project the centralized indices onto a 2D annulus. The angular position is determined by mixing spatial-origin angle and grid-index angle, and the radius $R$ controls scale.

(iii) *Target Plane Rotation*: rotate the 2D annulus into a plane in 3D space whose normal is aligned with the text-index direction, enforcing orthogonality between image-token geometry and text-token indexing.

In M-RoPE (Wang et al., 2024a), image token indices are represented by 2D width–height coordinates, while text tokens use the standard 1D position indices. As shown in Figure 3(a), we model the image token indices as a regular grid $C = \{(x_{ij}, y_{ij})\}_{i \in W, j \in H}$, where $W = \{0, 1, \ldots, w - 1\}$ and $H = \{0, 1, \ldots, h - 1\}$ denote the index sets along the width and height axes, respectively. Here, $w$ and $h$ correspond to the width and height of the tokenized image; we associate $W$ with the $x$-axis and $H$ with the $y$-axis. The goal of CIP is to transform the original grid indices $C$ into indices that are decoupled from text tokens, yielding $C_{\mathrm{proj}} = \{(x_{ij}^{\mathrm{proj}}, y_{ij}^{\mathrm{proj}})\}$. These projected indices are then used directly for RoPE computation.

### 4.1.1. COORDINATE CENTRALIZATION

To facilitate subsequent transformations, we first center the image token index coordinates. Specifically, the geometric center $P_{\mathrm{center}} \in \mathbb{R}^2$ of the image token indices is calculated as follows:

$$P_{\mathrm{center}} = \frac{1}{2}\left(\max_i(C_i) + \min_i(C_i)\right) \tag{5}$$

We then subtract this center point from all original coordinates to obtain the centered coordinates:

$$C' = C - P_{\mathrm{center}} \tag{6}$$

This ensures that the geometric center of $C' = \{(x_{ij}', y_{ij}')\}$ is located at the origin $(0, 0)$, providing a natural reference frame for subsequent projection and rotation. Since this step is a pure translation, it preserves all pairwise relative offsets within the image-token index set. With this normalized coordinate frame, we can now assign each token an annular angle in a consistent manner.

### 4.1.2. MIXED-ANGLE ANNULAR MAPPING

To construct a cone-like structure that effectively decouples the text token indices from the image token indices, we first transform the centered image token coordinates $C'$ into polar coordinates and project them onto a 2D annulus. During this transformation, the angular position of each point on the annulus is determined by a combination of its spatial-origin angle (SA) and grid-index angle (GA), while the radius $R$ remains flexible. The resulting 2D annular structure is illustrated in Figure 3(b). We detail the calculation of these two angles and the radius in the following.

**Angle Calculation:** We combine two complementary angles to balance spatial structure with index information, determining the transformed angle for each image token index:

(1) *Spatial-Origin Angle $\theta_{ij}^{SA}$ (SA)*: we first compute the polar angle of each centered point $(x_{ij}', y_{ij}')$:

$$\theta_{ij}^{\mathrm{atan2}} = \mathrm{atan2}(y_{ij}', x_{ij}') \tag{7}$$

where the function $\mathrm{atan2}(y, x)$ returns the angle between the point $(x, y)$ and the positive $x$-axis in $(-\pi, \pi]$. We then normalize these angles to the range $[0, 2\pi)$:

$$\theta_{\min} = \min_{i,j}(\theta_{ij}^{\mathrm{atan2}}), \quad \theta_{\max} = \max_{i,j}(\theta_{ij}^{\mathrm{atan2}}) \tag{8}$$

$$\Delta\theta = \theta_{\max} - \theta_{\min} \tag{9}$$

Thus, as illustrated in Figure 3(e), the SA is given by:

$$\theta_{ij}^{\mathrm{SA}} = \begin{cases} \frac{\theta_{ij}^{\mathrm{atan2}} - \theta_{\min}}{\Delta\theta} \times 2\pi & \text{if } \Delta\theta > 0 \\ 0 & \text{if } \Delta\theta \leq 0 \end{cases} \tag{10}$$

(2) *Grid-Index Angle $\theta_{ij}^{GA}$ (GA):* We flatten the $H \times W$ grid into a 1D sequence with $N = H \times W$ points, assigning each point a uniformly spaced angle based on its flattened index $k \in \{0, ..., N-1\}$:

$$\theta_k^{\mathrm{GA}} = \frac{k}{N} \times 2\pi \tag{11}$$

mapping the index $k$ back to the grid position $(i, j)$ yields $\theta_{ij}^{\mathrm{GA}}$, ensuring the angles are equally spaced around the annulus, as shown in Figure 3(d). Importantly, GA is deterministic rather than random: it fol-

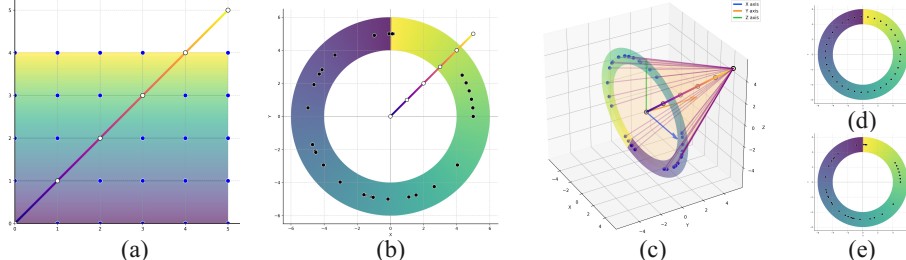

*Figure 3.* Transformation steps for *Circular Image Token Index Projection* (**CIP**): (i) coordinate centralization, (ii) mixed-angle annular mapping, and (iii) target-plane rotation (Sec. 4.1). For clarity, we align the starting points of text token indices and image token indices in the figure; this does not affect relative positional distances and is made without loss of generality. (a) Initial M-RoPE (Wang et al., 2024a) indices at step (i); (b) 2D annular structure after steps (i) and (ii); (c) 3D annular structure after step (iii); (d) grid-index angle (GA) in step (ii); (e) spatial-origin angle (SA) in step (ii).

lows the same raster-scan serialization used by ViT-style patch flattening, preserving a one-to-one correspondence between grid coordinates and circular angles.

(3) *Angle Mixing:* The final mixed angle $\theta_{ij}^{\mathrm{mix}}$ is computed by a weighted average of the two strategies:

$$\theta_{ij}^{\mathrm{mix}} = \alpha \cdot \theta_{ij}^{\mathrm{SA}} + (1 - \alpha) \cdot \theta_{ij}^{\mathrm{GA}} \qquad (12)$$

The coefficient $\alpha \in [0, 1]$ controls the balance between preserving spatial information and enhancing the uniqueness of each position. While SA retains more spatial structure, GA yields a clearer separation between positions, making it easier for the model to distinguish them. This is particularly important for avoiding *angular collapse*: tokens lying on similar rays from the image center can receive nearly identical SA angles, whereas GA regularizes the annular layout by spreading tokens uniformly.

**Radius Calculation:** The choice of radius $R$ determines the scale of the transformed coordinates, which in turn affects the effective frequency range of RoPE (Su et al., 2024) and the magnitude of intra-image relative offsets. We consider two practical strategies:

(1) Fixed: Use a predefined constant value $R_{\mathrm{fix}}$.

(2) Automatic (auto-$k$): Scale $R$ based on a measure of the spread of the centered coordinates $C'$, such as the maximum $L_2$ norm:

$$R_{\mathrm{auto}} = k \times \max_{i,j} \|(x'_{ij}, y'_{ij})\|_2 \qquad (13)$$

where $k$ is a predefined scaling factor (*e.g.*, $k = 1$ or $k = 2$).

**Mapping to the Annulus:** Based on the computed angle $\theta_{ij}^{\mathrm{mix}}$ and radius $R$, the new coordinates of each image token

index on the $XY$-plane are given by $x_{ij}^{\mathrm{circ}} = R\cos(\theta_{ij}^{\mathrm{mix}})$ and $y_{ij}^{\mathrm{circ}} = R\sin(\theta_{ij}^{\mathrm{mix}})$, which collectively form an annulus $C_{\mathrm{circ}} = \{(x_{ij}^{\mathrm{circ}}, y_{ij}^{\mathrm{circ}})\}$, as illustrated in Figure 3(b).

### 4.1.3. TARGET PLANE ROTATION

The purpose of this step is to rotate the annulus so that its plane is orthogonal to the text-index direction. Concretely, we align the normal vector of the annulus plane with the text position direction $V_{\mathrm{text}}$. Equivalently, the image annulus is rotated to face the text token in the RoPE index space. This orthogonality is the geometric condition behind our cone-like construction: for any fixed text token $t$, the relative offsets $\{m_i - m_t\}_{i \in I}$ have identical Euclidean norms, i.e., $\|m_i - m_t\|_2$ is constant over image tokens. Therefore, the geometry-dependent factor introduced by RoPE becomes uniform across $i$ for this query token, enforcing $\mathrm{PTD} = 0$ and eliminating geometry-dependent positional preference in cross-modal attention.

After the mixed-angle mapping, visual token index points lie on $C_{\mathrm{circ}}$ in the $XY$-plane. We extend them to 3D by setting the third coordinate to zero, and then map the annulus to the target plane via an orthonormal basis $\{\mathbf{u}, \mathbf{v}, \mathbf{n}\}$, where $\mathbf{n}$ is aligned with $V_{\mathrm{text}}$:

(1) **Define the normal.** Normalize the text direction to obtain the unit normal vector $\mathbf{n}$:

$$\mathbf{n} = \frac{V_{\mathrm{text}}}{\|V_{\mathrm{text}}\|_2} = (n_x, n_y, n_z) \qquad (14)$$

(2) **Construct an orthonormal basis.** Choose a unit vector $\mathbf{u}$ that is orthogonal to $\mathbf{n}$, and then define $\mathbf{v}$ by a cross product:

$$\mathbf{u}' = (-n_y, n_x, 0) \quad \mathbf{u} = \frac{\mathbf{u}'}{\|\mathbf{u}'\|_2}, \quad \mathbf{v} = \mathbf{n} \times \mathbf{u} \quad (15)$$

so that $\mathbf{u}, \mathbf{v}, \mathbf{n}$ forms a right-handed orthonormal frame.

*Table 2.* Performance of VLM Instruct models and our method (improvement over Qwen2.5-VL (SFT) shown in parentheses).

| Dataset | SAIL-VL 2B (Dong et al., 2025) | InternVL2.5 4B (Chen et al., 2024) | MiniCPM V-2 2.8B (Yao et al., 2024) | MiniCPM V-2.6 8B (Yao et al., 2024) | Phi-3.5 vision 4.2B (Abdin et al., 2024) | Qwen2.5-VL 3B (Bai et al., 2025) | Qwen2.5-VL (SFT) 3B | Ours 3B |
|---|---|---|---|---|---|---|---|---|
| MMMU$_\text{val}$ (Yue et al., 2024) | 41.44 | 51.56 | 37.00 | 43.44 | 44.44 | 50.22 | 51.56 | **52.11** (+0.55) |
| MMMU-Pro$_\text{overall}$ (Yue et al., 2025) | 14.51 | 26.65 | 14.77 | 20.26 | 16.42 | 27.92 | 28.01 | **28.44** (+0.43) |
| MathVista$_\text{mini}$ (Lu et al., 2024) | 60.70 | 60.60 | 40.80 | 60.20 | 43.70 | 62.40 | 62.40 | **63.40** (+1.00) |
| AI2D (Kembhavi et al., 2016) | 77.72 | 81.38 | 64.77 | 81.28 | 77.59 | 78.14 | 79.22 | **81.80** (+2.58) |
| RealWorldQA (X.AI, 2024) | 63.01 | 64.97 | 55.03 | 65.62 | 53.99 | 65.75 | 66.10 | **66.54** (+0.44) |
| **Avg Score** | 51.48 | 57.03 | 42.47 | 54.16 | 47.23 | 56.89 | 57.46 | **58.46** (+1.00) |

(3) **Coordinate transformation.** For each point $P_{ij}^\text{circ} = (x_{ij}^\text{circ}, y_{ij}^\text{circ}, 0)$, we map it onto the target plane by:

$$P_{ij}^\text{proj} = x_{ij}^\text{circ}\mathbf{u} + y_{ij}^\text{circ}\mathbf{v} \qquad (16)$$

yielding the projected set $C_\text{proj} = \{(x_{ij}^\text{proj}, y_{ij}^\text{proj}, z_{ij}^\text{proj})\}$ (Figure 3(c)).

Intuitively, this rotation makes the text direction the normal of the image annulus. Therefore, for any fixed text token, the geometric contribution induced by RoPE becomes uniform over image tokens, avoiding a geometry-dependent positional preference.

**PTD guarantee.** By construction, $C_\text{proj}$ lies on an annulus in a plane whose normal is aligned with $V_\text{text}$. This enforces that, for each text token index, the induced cross-modal RoPE distance to all image tokens is constant, and thus PTD = 0.

### 4.2. Alternating Geometry Encoding (AGE)

Although Circle-RoPE removes spurious cross-modal geometric bias by enforcing PTD = 0, a VLM must simultaneously support two types of structure: **(i) unbiased cross-modal alignment** (grounding text to relevant regions without positional interference) and **(ii) strong intra-image locality** (aggregating nearby visual features for fine-grained perception). M-RoPE naturally instills the latter through a grid-based, "convolution-like" inductive bias that favors short-range 2D interactions.

This leads to a key observation: the positional "bias" induced by grid locality is context-dependent. It is often beneficial for image-to-image attention (preserving intra-image structure), but can be detrimental for text-to-image alignment when image and text share a coupled index space. Pure Circle-RoPE, by construction, prioritizes cross-modal equidistance and thus implicitly relaxes strict 2D locality in image-to-image attention.

To reconcile these objectives without forcing a single geometry to serve incompatible roles, we introduce Alternating Geometry Encoding (AGE). AGE assigns different layers to emphasize distinct geometric priors: Circle-RoPE layers promote unbiased semantic grounding, while M-RoPE layers reinforce structured visual extraction.

Formally, AGE specifies a layer-wise schedule $s(\ell) \in \{\text{Circle-RoPE}, \text{M-RoPE}\}$ and applies the corresponding encoding at each Transformer layer $\ell$. In Sec. 5.5, we evaluate several practical schedules (all-Circle, lower-only, upper-only, and alternating), with additional discussion provided in Appendix B.6.

## 5. Experiments

In this section, we report (i) main results on the baseline model (Qwen2.5-VL) and other models in the unified benchmark suite, followed by (ii) ablations that isolate CIP and AGE, and then (iii) diagnostic/verification experiments that explain why Circle-RoPE helps (spatial grounding and decoupling). Finally, we provide generalizability evidence on a different architecture and discuss the adaptation cost in practice.

### 5.1. Training Setting

To evaluate the effectiveness of our method, we adopt Qwen2.5-VL (Bai et al., 2025) and LLaVA (Liu et al., 2023a) as baseline models. We only replace the positional encoding module; all other configurations follow the original implementations. During training, we update the parameters of the LLM component while keeping the vision encoder and the vision-language projection layers frozen. All experiments use a unified training setup, with full hyperparameter details provided in Table 11 in the appendix. For supervised fine-tuning, we randomly sample one-tenth of the MAmmoTH-VL Instruct dataset (**12M**) (Guo et al., 2025) and exclude all video data, resulting in a subset named MAmmoTH-VL-Sub (**1M**). Even with this reduced data size, Circle-RoPE achieves consistent improvements over the corresponding baselines.

### 5.2. Comparison with Other Models and the Baseline

Table 2 reports the main results on a unified set of multimodal benchmarks, including **(i)** a controlled comparison against our baseline (Qwen2.5-VL (SFT)) and **(ii)** a horizontal comparison with representative open-source VLMs. We emphasize the baseline comparison because Circle-RoPE modifies only the positional encoding while keeping the rest of the training recipe unchanged, making the performance

gains more attributable to the proposed method.

To ensure a fair comparison for the open-source models in Table 2, we evaluate all models with VLMEvalKit (Duan et al., 2024) under a unified protocol. Because the evaluation toolkit and the GPT version may differ from those used in the original papers, the reported numbers may not exactly match the official results.

### 5.3. Ablation on Circular Mapping (CIP)

We conducted ablation studies on the parameters used in Circular Image Token Index Projection (CIP). To validate the effectiveness of angle mixing and to select the optimal radius, we designed a series of ablation experiments. Specifically, we varied the angle mixing parameter $\alpha$ and explored different strategies for calculating the radius. As shown in Table 4, the model achieves the most balanced performance when $\alpha = 0.5$ and the radius is set to 10.

Additionally, we provide results for the baseline model after supervised fine-tuning (SFT) on the MAmmoTH-VL-Sub (**1M**) dataset. This allows for a direct comparison of how different parameter configurations affect model performance under the same conditions.

### 5.4. $\mathrm{PTD} = 0$ **Requires Preserved Image Geometry**

While enforcing $\mathrm{PTD} = 0$ removes cross-modal distance variance, it does not by itself guarantee better multimodal reasoning, as it may inadvertently collapse the intra-image spatial structure. To disentangle these effects, we compare two $\mathrm{PTD} = 0$ strategies in Table 5: (1) **Unordered** embedding, which collapses all image tokens to a single shared index; and (2) **Circle-RoPE**, which arranges image tokens on an annulus (a cone-like geometry w.r.t. the text axis) to keep cross-modal distances uniform while preserving visual geometry.

Despite also satisfying $\mathrm{PTD} = 0$, Unordered exhibits a clear degradation (Avg 54.67) relative to M-RoPE (Avg 56.76) and Circle-RoPE (Avg 57.76). This indicates that our gains do not come from "removing distance variance" alone: **preserving image-token geometry is crucial**. Circle-RoPE achieves this by keeping intra-image structure through the annular angular layout (and a controllable radius), while enforcing orthogonality to the text-index direction so that cross-modal RoPE-induced scaling is uniform for a fixed text token.

We further distinguish equidistance from simply increasing text–image separation. A naive large-offset concatenation preserves image structure but can push relative offsets into a regime where RoPE attenuates long-range interactions, weakening cross-modal alignment. In contrast, Circle-RoPE enforces equidistance (yielding an isotropic positional prior for text-to-image attention) without collapsing visual posi-

tions, so spatial reasoning and grounding remain intact.

### 5.5. Ablation on Alternating Geometry Encoding(AGE)

We evaluate whether layer-wise alternation between Circle-RoPE and M-RoPE (AGE; Sec. 4.2) improves performance. Our hypothesis is that AGE prevents the model from over-fitting to a single geometric view: by exposing the network to both cone-like (Circle-RoPE) and grid-like (M-RoPE) manifolds, AGE acts as a geometric regularizer.

To assess how the layer-wise schedule affects performance, we compare four configurations: (1) applying Circle-RoPE in all layers; (2) applying Circle-RoPE only in the upper layers (layers 19–36); (3) applying Circle-RoPE only in the lower layers (layers 1–18); and (4) applying AGE. We further discuss why alternating helps optimization and spatial reasoning in Appendix B.6.

As shown in Table 3, AGE yields the most robust performance among all configurations. We attribute this to the *structural diversity* introduced by alternating geometries: it encourages specialization across attention heads and layers, where Circle-RoPE layers facilitate unbiased cross-modal reasoning (e.g., grounding text to relevant regions) while M-RoPE layers maintain precise intra-image spatial awareness (e.g., chart reading and fine-grained layout). This complementary specialization helps AGE achieve a better balance across diverse benchmarks compared to using either encoding exclusively.

### 5.6. Diagnostics: Spatial Grounding and Decoupling

To verify that Circle-RoPE improves cross-modal decoupling without sacrificing spatial grounding, we evaluate on the TAM (Token Attention Mask) benchmark. TAM reports visual grounding precision (Obj-IoU), decoupling capability (Func-IoU), and the overall multimodal spatial mapping accuracy (F1-IoU).

*Table 6.* Performance on the TAM benchmark. Circle-RoPE demonstrates superior visual grounding (Obj-IoU) and significant improvements in decoupling capability (Func-IoU) compared to the baseline.

| Model | Obj-IoU | Func-IoU | F1-IoU |
| --- | --- | --- | --- |
| Qwen2.5-VL (SFT) | 26.15 | 71.19 | 38.25 |
| **Circle-RoPE** | **26.23** | **74.64** | **38.81** |

As shown in Table 6, Circle-RoPE slightly improves Obj-IoU, indicating that spatial grounding is preserved. More importantly, the gain in Func-IoU (+3.45) suggests that reducing spurious cross-modal positional bias helps the model better suppress irrelevant visual regions, leading to more accurate and robust spatial understanding.

We additionally provide attention-map visualizations in the

*Table 3.* Performance comparison across different AGE configurations.

| Method | Strategy | MMMU (val) | MMMU_Pro | MathVista_MINI | AI2D_TEST | RealWorldQA | Avg score |
|---|---|---|---|---|---|---|---|
| Qwen2.5-VL (SFT) | — | 51.56 | 28.01 | 62.40 | 79.22 | 66.10 | 57.46 |
| Circle-RoPE | 1 | 51.32 | 28.42 | **65.20** | 80.39 | 66.34 | 58.33 |
| Circle-RoPE | 2 | 52.66 | 28.51 | **65.20** | 79.80 | 65.94 | 58.42 |
| Circle-RoPE | 3 | **53.48** | **28.62** | 64.50 | 79.30 | 66.32 | 58.44 |
| Circle-RoPE | 4 | 52.11 | 28.44 | 63.40 | **81.80** | **66.54** | **58.46** |

*Table 4.* Performance comparison across different CIP configurations.

| $\alpha$ | Radius | MMMU$_{val}$ (Yue et al., 2024) | MMMU-Pro$_{overall}$ (Yue et al., 2025) | MathVista$_{mini}$ (Lu et al., 2024) | Avg Score |
|---|---|---|---|---|---|
| baseline | | 50.22 | 27.92 | 62.40 | 46.85 |
| $\alpha = 0$ | auto | **52.38** | 28.12 | 61.70 | 47.40 |
| $\alpha = 0$ | 5 | 51.32 | 29.01 | 62.40 | 47.58 |
| $\alpha = 0$ | 10 | 51.49 | **29.13** | 62.70 | 47.77 |
| $\alpha = 0.3$ | 10 | 52.05 | 28.50 | 63.30 | 47.95 |
| $\alpha = 0.5$ | 10 | 52.11 | 28.44 | **63.40** | **47.98** |
| $\alpha = 0.7$ | 10 | 52.03 | 28.39 | 62.90 | 47.77 |
| $\alpha = 1$ | 10 | 52.16 | 28.35 | **63.40** | 47.97 |
| $\alpha = 0.5$ | auto | 50.04 | 26.64 | 62.20 | 46.29 |

*Table 5.* Ablation of PTD=0 strategies. Circle-RoPE outperforms Unordered embedding, proving the necessity of preserving spatial structure.

| Strategy | MMMU | MMMU_Pro | MathVista | AI2D | RealWorldQA | Avg |
|---|---|---|---|---|---|---|
| M-RoPE (Baseline) | 50.56 | 27.51 | 61.40 | 78.22 | 66.10 | 56.76 |
| Unordered ($PTD = 0$) | 48.55 | 25.50 | 59.50 | 75.50 | 64.31 | 54.67 |
| **Circle-RoPE** ($PTD = 0$) | **51.11** | **27.94** | **62.40** | **80.80** | **66.54** | **57.76** |

appendix to qualitatively illustrate that Circle-RoPE concentrates more on question-relevant image regions.

### 5.7. Generalizability to Other Architectures

To evaluate the generalizability of Circle-RoPE, we conduct an ablation study on LLaVA (Liu et al., 2023a), whose architecture differs from the primary model used in this work. We adopt LLaVA-OneVision-Qwen2-0.5B as the base model and train/evaluate on MAmmoTH-VL-Sub.

We compare the following variants to isolate the effect of positional encoding: **LLaVA [1D-RoPE] (base)**: the original model with 1D-RoPE; **LLaVA [M-RoPE]**: replacing 1D-RoPE with M-RoPE from Qwen2.5-VL; **LLaVA [Circle-RoPE]**: replacing 1D-RoPE with Circle-RoPE.

Results are summarized in Table 7. Circle_RoPE achieves the best overall performance across all reported metrics.

*Table 7.* Ablation study on the LLaVA-0.5B model to verify the generalizability of Circle-RoPE. Our method achieves the best performance across all benchmarks, demonstrating its effectiveness on a different model architecture.

| Model | MMMU-val | MMMU_Pro-avg | MathVistamini | Avg Score |
|---|---|---|---|---|
| LLaVA [1D-RoPE] | 32.22 | 12.92 | 35.70 | 26.95 |
| LLaVA [M-RoPE] | 32.59 | 12.81 | 35.40 | 26.93 |
| **LLaVA [Circle-RoPE]** | **32.77** | **13.21** | **36.10** | **27.36** |

As shown in Table 7, Circle-RoPE outperforms both the LLaVA baseline and the M-RoPE variant, indicating that the benefits of Circle-RoPE generalize beyond the Qwen-VL family. For these experiments, we directly reuse the hyperparameters ($\alpha$ and $R$) selected on Qwen2.5-VL, without any architecture-specific tuning, yet still observe consistent gains.

## 6. Conclusion

We revisit the challenges of applying RoPE to multimodal VLMs, where coupled text–image indexing can introduce spurious positional bias. To quantify this effect, we propose the per-token distance (PTD) metric. Guided by PTD, we introduce Circle-RoPE, which preserves intra-image relative geometry while decoupling cross-modal positional relations, thereby improving cross-modal alignment and robustness.

## Acknowledgements

This work is funded by Peking University–BHP Carbon and Climate Wei-Ming PhD Scholars Program (Program Name: Research on Low-Carbon and Energy-Efficient Large Model Architectures; Program Number: WM202505).

## Impact Statement

This paper presents work whose goal is to advance the field of Machine Learning. There are many potential societal consequences of our work, none which we feel must be specifically highlighted here.

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

# Appendix

This appendix includes further analysis and discussion, related work, the hyperparameters adopted in our experiments, and pseudocode implementations.

## A. Formal Proof of Cross-Modal Bias Elimination

In this section, we formally connect our PTD metric to RoPE-induced geometric attention bias. We first derive the expected RoPE attention logit as an explicit function of the relative RoPE index offset, and then show that the magnitude of geometric bias is (up to constants) controlled by PTD. Throughout, we consider a fixed text query token $t$ and a set of image key tokens $i \in I$.

**RoPE attention logit as a rotation kernel.** With RoPE, the attention logit between a text query $\mathbf{q}_t$ at index $m_t$ and an image key $\mathbf{k}_i$ at index $m_i$ can be written as

$$s_{t,i} = (\mathcal{R}_{m_t} \mathbf{q}_t)^\top (\mathcal{R}_{m_i} \mathbf{k}_i) = \mathbf{q}_t^\top \mathcal{R}_{\delta_{t,i}} \mathbf{k}_i, \quad \delta_{t,i} \triangleq m_i - m_t. \tag{17}$$

For standard RoPE, $\mathcal{R}_\delta$ is block-diagonal with $2 \times 2$ rotation blocks. Let the embedding dimension be $d = 2H$. For $h = 1, \ldots, H$, denote the corresponding 2-D subvectors $\mathbf{q}_t^{(h)} \in \mathbb{R}^2$, $\mathbf{k}_i^{(h)} \in \mathbb{R}^2$ and rotation angle $\varphi_h(\delta)$. Then

$$s_{t,i} = \sum_{h=1}^{H} \left( \mathbf{q}_t^{(h)} \right)^\top \mathbf{R}(\varphi_h(\delta_{t,i})) \, \mathbf{k}_i^{(h)}, \tag{18}$$

where $\mathbf{R}(\theta) = \begin{bmatrix} \cos\theta & -\sin\theta \\ \sin\theta & \cos\theta \end{bmatrix}$.

**A semantic-conditioned expectation that decays with relative offset.** We now take expectation over the token representations while conditioning on *semantic relevance*. Formally, let $\mathbb{E}[\cdot \mid \text{sem}(t,i)]$ denote expectation conditioned on the semantic match between $t$ and $i$. Define the $2 \times 2$ cross-covariance (semantic alignment) for head/block $h$:

$$\mathbf{\Sigma}^{(h)} \triangleq \mathbb{E}\left[ \mathbf{k}_i^{(h)} \left( \mathbf{q}_t^{(h)} \right)^\top \mid \text{sem}(t,i) \right]. \tag{19}$$

Then the expected RoPE logit satisfies

$$\mathbb{E}[s_{t,i} \mid \text{sem}(t,i)] = \sum_{h=1}^{H} \mathbb{E}\left[ \left( \mathbf{q}_t^{(h)} \right)^\top \mathbf{R}(\varphi_h(\delta_{t,i})) \, \mathbf{k}_i^{(h)} \mid \text{sem}(t,i) \right] \tag{20}$$

$$= \sum_{h=1}^{H} \text{tr}\left( \mathbf{R}(\varphi_h(\delta_{t,i})) \, \mathbf{\Sigma}^{(h)} \right). \tag{21}$$

**Assumption (block-isotropic semantic alignment).** For semantically matched pairs, a common and empirically consistent approximation is that $\mathbf{\Sigma}^{(h)}$ is approximately isotropic in each 2D RoPE subspace:

$$\mathbf{\Sigma}^{(h)} \approx a_h \mathbf{I}_2, \quad a_h \geq 0. \tag{22}$$

Under equation 22, using $\text{tr}(\mathbf{R}(\theta)) = 2\cos\theta$, we get a closed form:

$$\mathbb{E}[s_{t,i} \mid \text{sem}(t,i)] \approx 2 \sum_{h=1}^{H} a_h \cos\big(\varphi_h(\delta_{t,i})\big). \tag{23}$$

**From the cosine sum to a distance-based decay factor.** Define $A \triangleq 2\sum_{h=1}^{H} a_h$ (a semantic scale, $A > 0$ for semantically relevant pairs), and the normalized *RoPE alignment kernel*

$$\mathcal{D}(\delta) \triangleq \frac{2\sum_{h=1}^{H} a_h \cos(\varphi_h(\delta))}{2\sum_{h=1}^{H} a_h} \in [-1, 1], \quad \mathcal{D}(0) = 1. \tag{24}$$

Then equation 23 becomes

$$\mathbb{E}[s_{t,i} \mid \text{sem}(t,i)] \approx A \cdot \mathcal{D}(\delta_{t,i}). \tag{25}$$

In many RoPE constructions (including multi-axis RoPE), the rotation angle satisfies $\varphi_h(\delta) = \langle \omega_h, \delta \rangle$ for some frequency vector $\omega_h$. When frequencies are spread across scales, the weighted cosine sum in equation 24 becomes increasingly oscillatory as $\|\delta\|_2$ grows, reducing coherent alignment; thus $\mathcal{D}(\delta)$ typically decreases with the offset magnitude. To match our PTD definition, we use the scalar proxy

$$r_{t,i} \triangleq \|\delta_{t,i}\|_2 = \|m_i - m_t\|_2, \quad \text{and write } \mathcal{D}(r_{t,i}) \text{ for } \mathcal{D}(\delta_{t,i}) \text{ on the relevant range.} \tag{26}$$

**Defining geometric attention bias (logit-level) and linking it to PTD.** For a fixed text token $t$, consider a subset of image tokens that are equally semantically relevant to $t$ (e.g., in the thought experiment where $\text{sem}(t,i)$ is fixed across $i$). Define the *geometric attention bias* (GAB) at the logit level as the mean absolute deviation of expected logits:

$$\text{GAB}_t \triangleq \frac{1}{N_{\text{image}}} \sum_{i \in I} |\mathbb{E}[s_{t,i} \mid \text{sem}] - \overline{s}_t|, \quad \overline{s}_t \triangleq \frac{1}{N_{\text{image}}} \sum_{i \in I} \mathbb{E}[s_{t,i} \mid \text{sem}]. \tag{27}$$

Using equation 25–equation 26, we have

$$\mathbb{E}[s_{t,i} \mid \text{sem}] \approx A \cdot \mathcal{D}(r_{t,i}), \quad \overline{s}_t \approx A \cdot \overline{\mathcal{D}}_t, \ \ \overline{\mathcal{D}}_t \triangleq \frac{1}{N_{\text{image}}} \sum_{i \in I} \mathcal{D}(r_{t,i}). \tag{28}$$

Therefore,

$$\text{GAB}_t \approx A \cdot \frac{1}{N_{\text{image}}} \sum_{i \in I} |\mathcal{D}(r_{t,i}) - \overline{\mathcal{D}}_t|. \tag{29}$$

Recall our PTD definition (mean absolute deviation of distances):

$$\bar{D}_t = \frac{1}{N_{\text{image}}} \sum_{i \in I} r_{t,i}, \quad \text{PTD} = \frac{1}{N_{\text{image}} N_{\text{text}}} \sum_{t \in T} \sum_{i \in I} |r_{t,i} - \bar{D}_t|. \tag{30}$$

**Theorem (PTD controls RoPE-induced geometric bias).** Assume that on the range of offsets encountered in a batch, $\mathcal{D}(r)$ is differentiable and monotone non-increasing, with slope bounded as

$$0 < \ell \leq -\mathcal{D}'(r) \leq L \quad \text{for all } r \in [r_{\min}, r_{\max}]. \tag{31}$$

Define the dataset-level bias as $\text{GAB} \triangleq \frac{1}{N_{\text{text}}} \sum_{t \in T} \text{GAB}_t$. Then, under equation 22 and equation 31,

$$A\ell \cdot \text{PTD} \lesssim \text{GAB} \lesssim AL \cdot \text{PTD}. \tag{32}$$

**Proof.** Fix $t$. By the mean value theorem and the slope bounds equation 31, for any $i \in I$,

$$\ell |r_{t,i} - \bar{D}_t| \leq |\mathcal{D}(r_{t,i}) - \mathcal{D}(\bar{D}_t)| \leq L |r_{t,i} - \bar{D}_t|. \tag{33}$$

Averaging equation 33 over $i \in I$ yields

$$\ell \cdot \frac{1}{N_{\text{image}}} \sum_{i \in I} |r_{t,i} - \bar{D}_t| \leq \frac{1}{N_{\text{image}}} \sum_{i \in I} |\mathcal{D}(r_{t,i}) - \mathcal{D}(\bar{D}_t)| \leq L \cdot \frac{1}{N_{\text{image}}} \sum_{i \in I} |r_{t,i} - \bar{D}_t|. \tag{34}$$

Since $\overline{\mathcal{D}}_t$ is an average of $\mathcal{D}(r_{t,i})$ and $\mathcal{D}(\bar{D}_t)$ is a constant, their difference changes only the centering but not the scaling; thus the same bounds hold (up to a negligible centering term) for $\frac{1}{N_{\text{image}}} \sum_i |\mathcal{D}(r_{t,i}) - \overline{\mathcal{D}}_t|$. Plugging into equation 29 gives

$$A\ell \cdot \frac{1}{N_{\text{image}}} \sum_{i \in I} |r_{t,i} - \bar{D}_t| \lesssim \text{GAB}_t \lesssim AL \cdot \frac{1}{N_{\text{image}}} \sum_{i \in I} |r_{t,i} - \bar{D}_t|. \tag{35}$$

Finally, averaging over $t \in T$ and using the PTD definition yields equation 32. $\qquad\square$

**Corollary 1 (Bias exists when** $\text{PTD} > 0$**).** If $\text{PTD} > 0$, then there exists at least one $t$ and two image tokens $i_a, i_b \in I$ such that $r_{t,i_a} \neq r_{t,i_b}$. By monotonicity of $\mathcal{D}$ and equation 25,

$$\mathbb{E}[s_{t,i_a} \mid \text{sem}] \neq \mathbb{E}[s_{t,i_b} \mid \text{sem}], \tag{36}$$

so purely geometric differences in RoPE indices induce unequal expected logits, i.e., geometric attention bias.

**Corollary 2 (**$\text{PTD} = 0$ **eliminates geometric bias and preserves softmax invariance).** If $\text{PTD} = 0$, then for every fixed text token $t$, all $r_{t,i}$ are identical across $i \in I$, hence $\mathcal{D}(r_{t,i})$ is constant and $\text{GAB}_t = 0$. Moreover, for the actual attention weights $\alpha_{t,i} = \exp(s_{t,i}) / \sum_j \exp(s_{t,j})$, if RoPE contributes only a uniform scalar shift/scale across all image tokens for a fixed $t$, then the softmax distribution over $i$ is unchanged (up to a temperature) and cannot encode a positional preference. This is exactly the invariance property exploited by our Circle-RoPE construction.

**Remark on necessity.** The above result should be interpreted as a statement about the geometric component of RoPE-induced cross-modal bias. For a fixed text token, if the RoPE-distance profile over image tokens is non-uniform, then there exist semantically matched image tokens whose expected logits differ purely because of their indices. Thus, eliminating this class of distance-induced geometric preference requires the cross-modal distance profile to be constant, which is exactly what $\text{PTD} = 0$ measures. Other non-orthogonal geometries could also satisfy this condition, but the orthogonal cone used by Circle-RoPE provides the simplest construction with a direct independence interpretation and efficient implementation.

# B. Additional Analysis

## B.1. Adaptation Cost

We instantiate Circle-RoPE on the architecturally closest backbone, *Qwen2.5-VL*, and monitor step-wise training dynamics under SFT. We observe that even a localized architectural change—replacing the positional encoding—can require substantial optimization for the model to adapt to the new positional distribution. We refer to this phenomenon as the *adaptation cost*.

*Table 8.* Step-wise training dynamics illustrating the *adaptation cost* when introducing Circle-RoPE on Qwen2.5-VL under SFT. At 3k steps, Circle-RoPE lags slightly behind; after ~8.5k steps it surpasses the baseline on MathVision. Best per column is in **bold**.

| Model | Step | Loss ↓ | MathVision ↑ |
|---|---|---|---|
| Qwen2.5-VL (SFT) | 3000 | 0.7997 | 20.16 |
| Circle-RoPE (SFT) | 3000 | 0.8077 | 20.13 |
| Qwen2.5-VL (SFT) | 8463 | **0.7666** | 20.56 |
| Circle-RoPE (SFT) | 8463 | 0.7725 | **20.95** |

Even on the closest backbone, Circle-RoPE exhibits a measurable *adaptation cost*: at 3k steps it slightly lags behind the SFT baseline (Table 8). With continued optimization (~8.5k steps), Circle-RoPE surpasses the baseline on MathVision (+0.39), suggesting that changing positional encoding requires non-trivial optimization to re-stabilize the representation geometry. From a practical standpoint, training the backbone *from scratch* is beyond our current compute budget; hence, we restrict ourselves to SFT-based adaptation. Despite this constraint, Circle-RoPE still delivers consistent and significant improvements across benchmarks, highlighting its practical effectiveness.

## B.2. Impact of Vision Encoder Fine-tuning

In our main experiments, we froze the vision encoder to prevent catastrophic forgetting and align with standard practices (e.g., LLaVA, BLIP-2). To empirically investigate whether this choice obscures potential degradation in visual representation quality or limits performance, we conducted an ablation study on the 3B model comparing the standard "Freeze ViT" setting against an "Unfreeze ViT" setting.

As shown in Table 9, unfreezing the ViT results in no significant negative impact on performance; in fact, it improves accuracy on simpler tasks like AI2D and MMMU (Val). This refutes the concern that Circle-RoPE might lead to inferior visual representations when the encoder is allowed to adapt.

*Table 9.* Performance comparison between Freezing and Unfreezing the Vision Encoder (ViT). Results show that unfreezing the ViT yields comparable or slightly better performance, refuting concerns about representation degradation.

| Setting | Steps | AI2D | MMMU-Pro (Vis) | MMMU-Pro (10c) | MMMU (Val) |
|---|---|---|---|---|---|
| Freeze ViT | 800 | 75.78 | **18.90** | **29.94** | 42.67 |
| Unfreeze ViT | 800 | **77.49** | 17.80 | 29.36 | **43.89** |

### B.3. Extension to N-Dimensions

Theoretically, our Circle-RoPE method scales to $n$-dimensional inputs by projecting them into an $(n + 1)$-dimensional space to ensure orthogonal decoupling from the text axis. For an input with $n$ spatial or temporal dimensions, the projection would map indices onto a hypersphere in $(n + 1)$-dimensional space.

In this work, we focused specifically on the $n = 2$ (Image) scenario for two primary reasons:

1. **Geometric Intuition and Visualization:** Mapping 2D image coordinates into a 3D space allows for clear visualization of the "cone-like" structure and the decoupling process (as illustrated in Figure 2). This interpretability was crucial for establishing the theoretical foundation.

2. **Foundational Validation:** Validating the theory on the standard image modality demonstrates the strong effectiveness of the proposed decoupling mechanism.

Dealing with $n > 2$ (e.g., spatio-temporal video data, where $n = 3$) requires mapping to spaces with dimensions $> 3$, which is an exciting direction for future work.

### B.4. Scope and Limitations

Our current experiments focus on static 2D image understanding. Circle-RoPE is geometrically extensible to higher-dimensional inputs such as videos by mapping spatio-temporal indices onto a hyperspherical manifold orthogonal to the text axis, but we leave large-scale 3D/video evaluation to future work. For multi-image inputs, we distinguish within-image spatial bias from inter-image order: Circle-RoPE enforces per-image text-to-patch equidistance, while image order can still be encoded by translating each image center along a shared temporal axis (Appendix B.7). This intentionally preserves sequence order between images and should not be interpreted as eliminating all temporal or inter-image positional information.

### B.5. Robustness to Different Image Resolutions

To verify whether Circle-RoPE is robust to varying input resolutions, we analyzed the performance on the AI2D dataset, whose samples exhibit significant variance in aspect ratio and resolution. The distribution of image sizes in the AI2D test set is shown in Table 10.

*Table 10.* Distribution of Image Resolutions in the AI2D Dataset.

| Image Resolution Range | Dataset Proportion (%) |
|---|---|
| $116 \sim 355$ | 4.3 |
| $255 \sim 394$ | 15.1 |
| $394 \sim 533$ | 23.9 |
| $533 \sim 672$ | 27.3 |
| $672 \sim 811$ | 10.4 |
| $811 \sim 1500$ | 19.0 |

Despite this high variance, our method achieves a significant improvement on AI2D (+2.58) compared to the SFT baseline (as shown in Table 2). This confirms that Circle-RoPE effectively handles diverse resolutions without degrading performance.

## B.6. Effectiveness of Alternating Geometry Encoding (AGE)

We introduce Alternating Geometry Encoding (AGE) into our method primarily for the following reasons:

**(1) Complementary strengths and preservation of spatial information.** While Circle-RoPE achieves image–text decoupling, it inevitably alters the strong grid-based spatial prior of image patches provided by the original RoPE. This prior refers to the explicit Cartesian row–column alignment induced by ViT-style patchification: neighboring grid coordinates correspond directly to local image translations. Mapping these coordinates to a circle preserves relative visual structure but converts the original linear grid into angular relations. By alternating the two encoding methods, the model benefits from both: it reduces cross-modal positional bias (from Circle-RoPE) and fully utilizes the fine-grained internal spatial structure of the image (from RoPE), achieving a "1+1>2" effect.

**(2) Compatibility with pre-trained knowledge and smooth transition.** Our models are fine-tuned from Qwen2.5-VL, whose weights are deeply adapted to the original RoPE. Compared with applying a completely new encoding scheme to one contiguous part of the network, an alternating strategy minimizes the "shock" to the existing weight distribution. This enables smoother and more data-efficient convergence under limited SFT data, better integrating the pre-trained knowledge with the new capabilities introduced by Circle-RoPE.

In summary, AGE serves as an optional but effective mechanism that (i) fuses complementary geometric biases to preserve spatial reasoning while reducing cross-modal positional bias, and (ii) eases optimization by providing a gentler transition from RoPE-adapted weights to Circle-RoPE-enhanced representations. Empirically, our ablations reflect these stability and performance benefits.

## B.7. Encoding Temporal Order in Multi-Image Sequences

When the input contains multiple images, we explicitly encode their sequential order by translating each image's annular-encoding center along a fixed global axis. This design separates two notions of positional information: Circle-RoPE removes spurious within-image patch preference for a fixed text query, whereas inter-image order is a real part of the prompt sequence that should be preserved. Concretely, let $c_i$ denote the center of the annular positional encoding for the $i$-th image in the sequence (indexed from $i=1$). We define a constant direction vector $g = [1, 1, 1]^\top$ and a stride $\Delta=1$ (default), and set

$$c_i^{\text{final}} \;=\; c_i \;+\; (i-1)\,\Delta\,g.$$

This translation assigns each image a unique location in the 3D positional space while keeping the within-image geometric structure determined by Circle-RoPE intact. In other words, Circle-RoPE targets equal text-to-patch distances *within each image* (per-image PTD = 0), while intentionally allowing different images to occupy different centers so that the model can distinguish images in a sequence.

For example, when we have a sequence with three images image1, image2, image3 whose original centers are at $0$, the final centers become

$$c_1^{\text{final}} = 0 + [0, 0, 0], \quad c_2^{\text{final}} = 0 + [1, 1, 1], \quad c_3^{\text{final}} = 0 + [2, 2, 2].$$

# C. Hyperparameters

*Table 11.* Training Hyperparameter Configuration for our method.

| Hyperparameter | Value |
|---|---|
| Base Model | Qwen2.5-VL-3B |
| Image Resolution | 512×512 |
| Global Batch Size | 128 |
| Learning Rate | 1e-6 |
| Optimizer | AdamW |
| LR Schedule | Cosine Decay |
| Number of Epochs | 1 |
| Warmup Ratio | 0.1 |
| Max Sequence Length | 4096 |

## D. Pseudocode implementation of Circle-RoPE

```python
import torch

def annular_image_token_projection(C: torch.Tensor, alpha: float, R: float, V_text: torch.Tensor):
    """
    Annular Image Token Projection in PyTorch style.

    Args:
        C (torch.Tensor): Original image token grid coordinates (N, 2).
        alpha (float): Angle mixing weight.
        R (float): Annulus radius.
        V_text (torch.Tensor): Text vector direction, shape (3,).

    Returns:
        torch.Tensor: Projected coordinates (N, 3).
    """

    # ==========================================================
    # Step 1: Coordinate Centralization
    # ==========================================================
    P_center = 0.5 * (C.max(dim=0).values + C.min(dim=0).values)   # (2,)
    C_prime = C - P_center                                         # (N, 2)

    # ==========================================================
    # Step 2: Mixed-Angle Annular Mapping
    # ==========================================================

    # 2a. Calculate Spatial-Origin Angle (SA)
    raw_angles = torch.atan2(C_prime[:, 1], C_prime[:, 0])        # (N,)
    min_angle = raw_angles.min()
    max_angle = raw_angles.max()
    delta_theta = max_angle - min_angle

    if delta_theta > 0:
        theta_SA = (raw_angles - min_angle) / delta_theta * 2 * torch.pi
    else:
        theta_SA = torch.zeros_like(raw_angles)

    # 2b. Calculate Grid-Index Angle (GA)
    N = C.shape[0]
    k = torch.arange(N, device=C.device)                          # (N,)
    theta_GA = (k.float() / N) * 2 * torch.pi

    # 2c. Mix Angles
    theta_mix = alpha * theta_SA + (1 - alpha) * theta_GA

    # 2d. Map to 2D annulus and expand to 3D
    x_circ = R * torch.cos(theta_mix)
    y_circ = R * torch.sin(theta_mix)
    C_circ = torch.stack([x_circ, y_circ, torch.zeros_like(x_circ)], dim=-1)  # (N, 3)

    # ==========================================================
    # Step 3: Target Plane Rotation
    # ==========================================================

    # 3a. Construct orthonormal basis from text vector
    n = V_text / V_text.norm()                                    # (3,)
    u_prime = torch.tensor([-n[1], n[0], 0.0], device=C.device)
    if u_prime.norm() < 1e-6:
        u_prime = torch.tensor([1.0, 0.0, 0.0], device=C.device)
    u = u_prime / u_prime.norm()
    v = torch.cross(n, u)

    # 3b. Project points from 2D annulus to 3D target plane
```

```
64      #     This is a linear combination of basis vectors u and v.
65      C_proj = C_circ[:, 0].unsqueeze(-1) * u + C_circ[:, 1].unsqueeze(-1) * v  # (N, 3)
66
67      return C_proj
```

## E. Visualization of Attention Map

To further evaluate the impact of our proposed method, we provide the visualization of attention distributions. The proposed methodology enables the visualization of cross-modal attention for Circle-RoPE and Qwen2.5-VL-3B-Instruct (Bai et al., 2025), with evaluations performed on the MMMU$_{\text{val}}$ benchmark (Yue et al., 2024). Concretely, we first isolate and extract the attention matrix from the final decoder layer. The average attention from all text tokens to their corresponding image regions is then computed, projected back to the image domain, and reconstructed into a coarse-grained grid. This grid is subsequently transformed into a heatmap, followed by smoothing and enlargement through bilinear interpolation. Finally, a power-law contrast enhancement is applied to highlight salient points. The visualization results show that our method is able to concentrate more effectively on the regions relevant to the given question while exhibiting fewer attentional allocations to irrelevant areas.

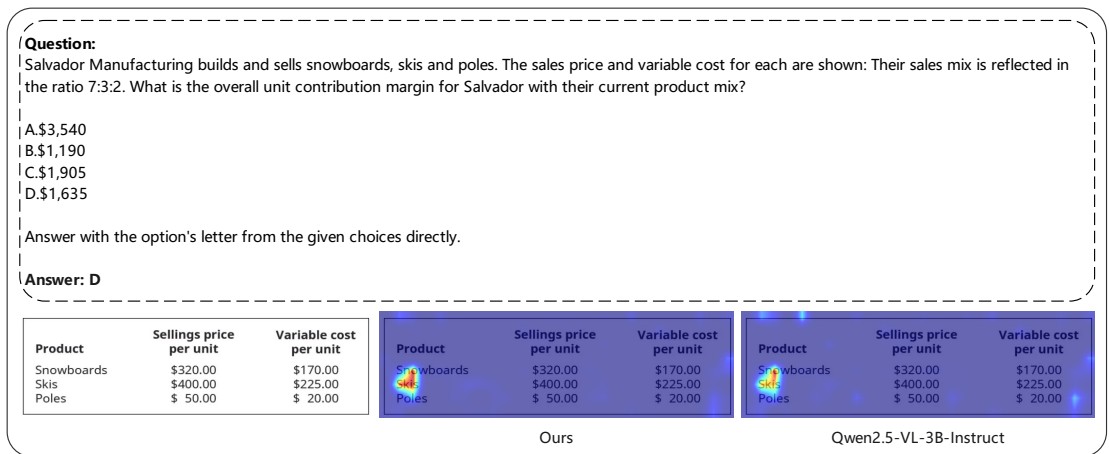

*Figure 4.* Attention visualization on MMMU$_{\text{val}}$: Circle-RoPE concentrates attention more on question-relevant regions compared to the baseline.

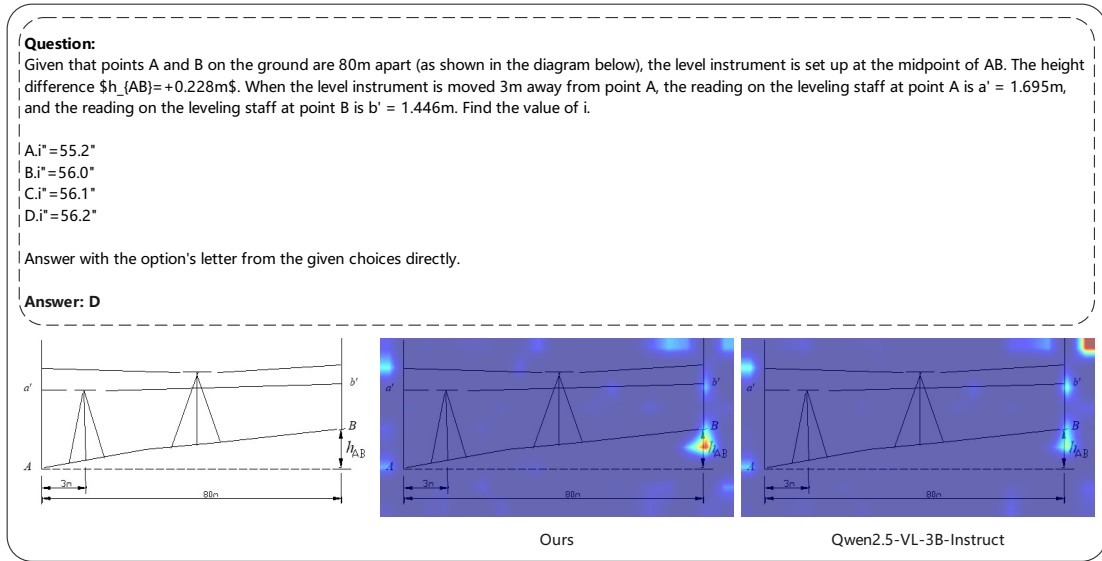

*Figure 5.* Additional attention visualization examples on MMMU$_{\text{val}}$, illustrating reduced attention to irrelevant regions with Circle-RoPE.

