# OpenReview forum: "Circle-RoPE: Cone-like Decoupled Rotary Positional Embedding for Vision-Language Models"
_ICML.cc/2026/Conference — ICML 2026 regular_

### Official Review · Reviewer_yjbo · 2026-03-08

**Soundness:** 2
**Presentation:** 3
**Significance:** 3
**Originality:** 3
**Overall Recommendation:** 3
**Confidence:** 4

**Summary:**

This paper studies a practical failure mode when extending Rotary Position Embedding to vision-language models: image tokens and text tokens share the same 1D index space, which can introduce a geometry-induced, semantics-irrelevant attention bias across modalities. The authors propose Per-Token Distance (PTD) to quantify whether each text token has an equal “distance” to all image tokens under the RoPE-induced geometry, and provide a formal analysis arguing that PTD = 0 is sufficient to eliminate the RoPE-induced geometric bias in cross-modal attention.
To achieve PTD = 0 while preserving intra-image spatial structure, the paper introduces Circle-RoPE, which uses Circular Image Token Index Projection (CIP) to map 2D image grid coordinates onto a circular plane orthogonal to the text position axis, yielding a “cone-like” geometry where each text token is equidistant to all image tokens. In addition, the paper proposes Alternating Geometry Encoding (AGE) that alternates Circle-RoPE and M-RoPE across layers to balance cross-modal decoupling and intra-image locality. Experiments on Qwen2.5-VL and other VLMs across multiple benchmarks show consistent improvements.

**Compliance With Llm Reviewing Policy:**

Affirmed.

**Key Questions For Authors:**

1.	Hyperparameter and resolution robustness: How sensitive are CIP’s key hyperparameters (e.g., α and R) to image resolution, patch size, and different backbones? Could you provide guidelines or an adaptive strategy?
2.	Mechanistic validation of AGE: Could you provide mechanistic evidence that AGE specifically restores/improves intra-image (I->I) spatial modeling, rather than acting as an empirical regularizer? Is there a more principled alternative to layer alternation that can achieve both T->I debiasing and I->I grid geometry?
3.	What structure is preserved: What exact notion of “image structure” does Circle-RoPE preserve, and can it be quantified? For example, do you preserve local adjacency, global topology, or distance correlations between the original 2D grid and the induced geometry?
4.	Failure modes and scope: Where does Circle-RoPE regress, and what are the typical failure modes? Could you provide task-family breakdowns and qualitative examples?

**Limitations:**

no.
I do not think the current limitations/impact discussion is adequate. At minimum, the paper should explicitly discuss:
1.	Applicability boundaries: behavior under task families, resolutions, and very long sequences.
2.	Hyperparameter dependence: transferability of α/R and AGE schedules.
3.	Impact on image geometry: Circle-RoPE’s ring projection may distort intra-image spatial structure, potentially hurting tasks that rely on fine spatial reasoning. The paper should further analyze and discuss this effect.

**Strengths And Weaknesses:**

Strengths
1.	Clear motivation with a measurable target. The cross-modal coupling issue in RoPE-based VLMs is well-motivated, and PTD provides a concrete quantity to diagnose and optimize for, rather than relying purely on qualitative attention visualizations.
2.	Theory-to-method linkage. The paper derives the RoPE attention logit structure and connects the magnitude of the geometry-induced bias to PTD, with the key claim that PTD = 0 removes the geometric bias. This provides a principled foundation for the design of Circle-RoPE.
3.	Low-intrusion, potentially broadly applicable change. The approach mainly changes the positional encoding / indexing scheme and appears easy to integrate into existing RoPE-based VLMs without architectural overhaul.
4.	Reasonable experimental coverage and ablations. The paper includes multiple benchmarks, plus ablations comparing different PTD=0 strategies and adding AGE, as well as TAM diagnostics that more directly target the coupling phenomenon.
Weaknesses
1.	Potential degradation of intra-image (I->I) spatial inductive bias. Although the paper effectively targets cross-modal positional bias, which is beneficial for T->I alignment, it may systematically weaken intra-image locality and other grid-like inductive biases in I->I attention. The proposed Alternating Geometry Encoding (AGE) is introduced to mitigate this issue, but the design appears somewhat empirically motivated. It would strengthen the paper to provide deeper mechanistic validation of why/when alternation works, or is there a better way?
2.	“Structure preservation” is asserted but not characterized. The comparison with Unordered embedding is helpful. However, while the paper claims Circle-RoPE “preserves image structure,” it remains unclear what kind of structure is actually preserved (e.g., local adjacency, global topology, relative distances) and why the ring construction retains that structure. A more explicit discussion would make the method’s claims more convincing and interpretable.
3.	Insufficient limitations and failure-mode analysis. Given the geometric re-parameterization, it would be valuable to identify scenarios where Circle-RoPE degrades performance. For example, tasks requiring strong local spatial reasoning, fine-grained localization, OCR-like sensitivity, or long prompts. A concrete failure analysis would improve the paper’s credibility and practical guidance.

---

> ### Author Rebuttal · Authors · 2026-03-31
>
> We thank the reviewer for this systematic and theoretically grounded evaluation. We respond below.
>
> > **W1 / Q2** — AGE mechanistic evidence
>
> AGE is not a generic regularizer. It alternates two operators with different roles: M-RoPE preserves grid-sensitive $I \to I$ geometry, while Circle-RoPE removes query-internal $T \to I$ bias by making each text token equidistant to the patches of one image. To test whether this is what the model actually does, we measured four layer-wise attention statistics on the 32 AGE layers:
>
> |Metric|Circle-M|
> |-|-|
> |$I \to I$ locality|+0.0148|
> |$T \to I$ uniformity|+0.000018|
> |$T \to I$ entropy|+0.0036|
> |$T \to I$ CV|-0.0051|
>
> $I \to I$ locality = fraction of image attention within 2D neighbors; $T \to I$ uniformity/entropy/CV measure text-to-image attention uniformity via negative log variance, normalized entropy, and coefficient of variation. In raw terms, locality is 0.1614 for Circle layers versus 0.1466 for M-RoPE layers, so $I \to I$ modeling does not degrade.
>
> |Strategy|All-M|All-Circle|Upper-only|Lower-only|AGE|
> |-|-|-|-|-|-|
> |Avg|57.46|58.33|58.42|58.44|58.46|
>
> The task pattern is also interpretable. Lower-only Circle helps early multimodal cleanup; upper-only Circle helps later semantic reasoning; AGE is best overall because the residual stream alternates grid-preserving and debiased representations across depth. We therefore view AGE as a principled geometric decomposition rather than an empirical trick.
>
> We also considered head-wise mixing but rejected it: contradictory positional signals from different heads would force $W_O$ to fuse inconsistent geometry, causing gradient conflict and breaking RoPE's rotation group structure (**see Reviewer EzGo, W3/Q4 for the full four-point argument**).
>
> > **W2 / Q3** — Structure preservation
>
> We agree with the reviewer that this phrase was too strong. The precise claim is that Circle-RoPE preserves relative angular ordering and part of local patch adjacency, while sacrificing radial distance. We quantify this with multi-scale $kNN@k$, the overlap between a patch's $k$ nearest neighbors before and after projection. **Full $5\alpha \times 6$ scale table in Reviewer jjCq (W1)**; concise comparison:
>
> |Method|k4|k16|k64|PTD|
> |-|-|-|-|-|
> |M-RoPE|1.000|1.000|1.000|0.64|
> |Circle ($\alpha=0.5$)|0.167|0.360|0.624|0|
> |Unordered|0|0|0|0|
>
> So Circle-RoPE does lose strict 2D Euclidean fidelity, but it preserves non-trivial neighborhood structure, especially at medium and larger scales, and this retained structure matters empirically: Table 5 gives 57.76 for Circle versus 54.67 for Unordered even though both have $\text{PTD}=0$. The ring preserves enough geometry while removing the specific cross-modal bias we target.
>
> > **W3 / Q4** — Failure modes
>
> The main failure mode is not spatial reasoning in general, but OCR-like chart reading:
>
> |Subset|Delta|
> |-|-|
> |logical reasoning|+8.11|
> |math word problem|+4.30|
> |arithmetic reasoning|+2.83|
> |numeric commonsense|+1.39|
> |visual QA|+1.12|
> |geometry reasoning|+0.84|
> |algebraic reasoning|+0.71|
> |textbook QA|+0.63|
> |geometry problem solving|+0.48|
> |figure QA|-0.74|
> |statistical reasoning|-1.99|
> |scientific reasoning|-2.46|
>
> ChartQA also drops slightly on both splits: test_human -1.20, test_augmented -0.64, overall -0.92. By contrast, geometry reasoning and geometry problem solving still improve. So the observed weakness is not "anything spatial," but tasks dominated by chart/OCR-style fine positional reading. That matches the geometry: Circle-RoPE removes spurious $T \to I$ preference but sacrifices exact radial separability. Scientific reasoning (-2.46, 122 samples from AI2D/ScienceQA/TQA) tests structural relations in diagrams that may depend on fine local directional topology — an open direction we flag in the revision.
>
> Two further theoretical risks, not yet observed: (1) very long prompts, because inter-query distance variation is intentionally not removed; (2) very small objects (<4 patches), where angular resolution on the ring may be insufficient. We add both as scope boundaries.
>
> > **Q1 / L1-L3** — Hyperparameters & limitations
>
> For hyperparameters, **see Reviewer EzGo (W2/Q2)**: all $\alpha/R$ configurations stay in a narrow band above baseline, no resolution bucket degrades, and $\alpha=0.5, R=10$ transfers to LLaVA without retuning (Table 7). Table 4 shows $\alpha$ in $[0.3, 0.7]$ yields <0.6pt Avg variation; $R$ sensitivity is even lower (range 0.37). MMMU-Pro favors low $\alpha$ (local adjacency) while MathVista favors high $\alpha$ (global layout); $\alpha=0.5$ balances both.
>
> Since RoPE rotations contain no learnable parameters, the geometric bias is structural; TAM Func-IoU +3.45 over the fully pretrained Qwen2.5-VL confirms it persists even after large-scale training (**see Reviewer jjCq, W3 for the full learning curve**).
>
> We fully agree the paper needs a clearer limitations section — **our comprehensive response to this concern is in our reply to Reviewer jKyy (L1)**.

---

> > ### Author Rebuttal · Reviewer_yjbo · 2026-04-07
> >
> > Thanks for author's effort for the rebuttal.

---

### Official Review · Reviewer_jjCq · 2026-03-12

**Soundness:** 3
**Presentation:** 3
**Significance:** 3
**Originality:** 3
**Overall Recommendation:** 5
**Confidence:** 4

**Summary:**

The paper points out the problem of spurious cross-modal positional bias in VLMs that use RoPE. The issue is that when text and visual tokens are indexed in a linked or shared way, the distances from text tokens to different image tokens become uneven, even though there is no real semantic reason for that. To study this effect, the authors introduce a metric called Per-Token Distance (PTD), which measures how uniform the distances are from a text token to a set of visual tokens in the positional space. Based on this idea, the paper proposes Circle-RoPE, which maps 2D image-token indices onto a ring that is orthogonal to the axis of text positions, forming a kind of cone-shaped geometry. The intuition is that this should make all visual tokens equally distant from each text token. On top of that, the authors also suggest AGE, which alternates Circle-RoPE and M-RoPE across layers.

**Compliance With Llm Reviewing Policy:**

Affirmed.

**Final Justification:**

I appreciate the authors’ careful and constructive rebuttal. After considering the rebuttal and the discussion, I remain positive about the paper. The response addressed my main concerns by clarifying the scope of the PTD result, making the image-structure claims more precise, and adding useful quantitative analysis on failure modes and learning dynamics. The absence of full pretraining experiments remains an important limitation, so I do not think the method’s behavior under full pretraining is fully settled yet. Still, within the scope claimed in the paper, I find the contribution technically sound, conceptually clear, and practically useful. I therefore maintain my score of 5.

**Key Questions For Authors:**

- The main results were obtained with SFT on 1M examples (e.g. one-tenth of the MAmmoTH-VLInstruct dataset(12M)). Do you have any (even preliminary) evidence on how Circle-RoPE behaves under more large-scale training? Do the learning curves suggest that the gap relative to the base model grows or shrinks as the amount of data increases? This feels pretty critical for judging the practical significance.

- Could you also quantify more clearly whether Circle-RoPE really preserves useful image geometry, rather than just removing some part of the cross-modal bias at the cost of distorting the underlying 2D structure? Are there any tasks or scenarios where Circle-RoPE actually hurts performance compared to M-RoPE?

- In the code, there is a parameter dff_rate that seems to interpolate between Circle-RoPE and standard coordinates. What exactly is its role? Is it actually used in the experiments?

**Limitations:**

- The fact that Circle-RoPE does not preserve the 2D geometry of the image in any strict sense;

- The spurious positional bias is an indexing artifact, but a trained model may compensate for it through the attention weights. If, during full pre-training, the model learns to ignore this positional bias, then Circle-RoPE is basically solving a problem that may not even exist.

- What’s really needed are experiments showing that this bias actually affects final performance under large-scale training

**Strengths And Weaknesses:**

Strengths:
- Important and underexplored problem.
The paper raises a meaningful question: in multimodal models, positional indices may indeed introduce an artificial bias into cross-modal attention. This is not a trivial issue, and it feels like something the community has not studied carefully enough yet. The problem setup is clear, and the motivation is easy to follow.

- Clear intuition and elegant geometric framing.
One of the strongest aspects of the paper is the intuition behind it. Looking at the issue through the geometry of positional indices, rather than only through the implementation details of RoPE, feels fresh and conceptually clean. Circle-RoPE is also a simple idea, easy to remember, and easy to explain, which is usually a good sign.

- PTD is a useful diagnostic lens.
PTD seems like a simple and intuitive way to quantify cross-modal positional separation. Even if the theoretical claims around it may need to be toned down, the metric itself still looks useful as an analytical tool. In particular, measuring something like inter-modal “distance uniformity” could be valuable beyond just this specific method.

- Practicality and reasonable empirical coverage.
The approach appears practical since it does not require redesigning the whole architecture and, according to the authors, only modifies the positional encoding. That makes the contribution potentially usable in practice. It is also good to see ablations on CIP, the AGE variant, and at least some attempt to test transfer to another architecture. The appendix discussion of adaptation cost and freeze/unfreeze choices also adds some honesty about limitations.

Weaknesses:
- The claim about preserving the internal geometry of the image is questionable.
If visual tokens are mapped onto a ring with a shared radius (R), then the original 2D grid is effectively projected onto a 1D manifold (the circle), and radial information is largely lost. Because of that, wording such as “preserving the internal spatial structure of the image” feels overstated. At best, the method preserves some relative ordering or angular structure, but not the full spatial geometry.

- PTD only captures part of the positional issue.
PTD controls the spread over image tokens for a fixed text token, but that does not fully solve the problem. In particular, the distance from a text token to the visual ring still depends on the text token’s own position. So later text tokens may remain farther from the visual part than earlier ones. In other words, the method may reduce one kind of cross-modal positional bias while still leaving another one in place.

- Limited training regime weakens the scope of the conclusions.
The authors are transparent that training from scratch is outside their compute budget, which is fair, but it still seriously limits what can be concluded. At the moment, it is unclear whether the benefits of Circle-RoPE would persist under full pre-training rather than adaptation-only settings. So the current results are encouraging, but they do not yet establish that the method is broadly superior in the regime that matters most.

---

> ### Author Rebuttal · Authors · 2026-03-31
>
> We are grateful for the reviewer's close reading and penetrating theoretical analysis. These comments have substantially sharpened our presentation. We respond.
>
> > **W1 / Q2 / L1** — Structure preservation
>
> The reviewer's correction is well-taken: Circle-RoPE does not preserve the full 2D grid in a strict Euclidean sense, and our original wording was indeed too strong. In the revision we replace "preserving the internal spatial structure" with "preserving relative angular ordering and part of local patch adjacency," and clarify that CIP is topologically a 1D manifold embedded in 3D RoPE index space.
>
> To quantify what is preserved, we measure multi-scale $kNN@k$ on a $16 \times 16$ grid. For each patch, $kNN@k$ is the fraction of its $k$ nearest neighbors in the original 2D grid that remain among its $k$ nearest neighbors after projection.
>
> |$\alpha$|k2|k4|k8|k16|k32|k64|
> |-|-|-|-|-|-|-|
> |0.00|0.500|0.471|0.275|0.242|0.385|0.503|
> |0.25|0.225|0.329|0.258|0.280|0.417|0.548|
> |0.50|0.098|0.167|0.215|0.360|0.496|0.624|
> |0.75|0.088|0.180|0.292|0.462|0.621|0.769|
> |1.00|0.100|0.201|0.383|0.524|0.683|0.832|
>
> The pattern reveals a fundamental 2D-to-ring tradeoff. Low $\alpha$ keeps raster-style local neighbors; high $\alpha$ keeps global angular layout. The crossover occurs at $k \approx H/2$ (≈7 for $16 \times 16$), consistent across all tested resolutions from $7 \times 7$ to $32 \times 32$, reflecting a topological constraint inherent to projecting a 2D grid onto a 1D manifold. $\alpha=0.5$ is not best at any single scale, but it is the only setting that avoids catastrophic failure at both ends, which is why it gives the best average performance in Table 4. We use kNN instead of a single global distance correlation because attention is driven most directly by local positional neighborhoods.
>
> |Method|k4|k16|k64|PTD|
> |-|-|-|-|-|
> |M-RoPE|1.000|1.000|1.000|0.64|
> |Circle ($\alpha=0.5$)|0.167|0.360|0.624|0|
> |Unordered|0|0|0|0|
>
> This is also why Circle-RoPE is substantially better than Unordered although both achieve $\text{PTD}=0$: Table 5 gives 57.76 vs. 54.67 (+3.09). So the gain is not "PTD=0 at any cost"; it depends on removing cross-modal bias while still preserving useful neighborhood information.
>
> > **W2** — Scope of PTD
>
> The reviewer raises an excellent point. We agree PTD captures only one component of the positional issue. The theorem is about query-internal bias: for a fixed text token $t$, unequal distances to image tokens insert a geometry-dependent factor into $s_{t,i}=q_t^T R_{m_i-m_t} k_i$. If two image tokens are equally relevant but sit at different relative distances, RoPE can prefer one purely because of geometry. When $\text{PTD}=0$, all those distances are equal, so the positional factor collapses to one constant scalar $\lambda$; after softmax, $\lambda$ changes only temperature, not which patch is favored. This proves $\text{PTD}=0$ is sufficient to remove query-internal cross-modal bias.
>
> It does not prove that every text token is globally equidistant to the visual block, and we do not claim that. We intentionally preserve this inter-query variation: in multi-image inputs, different text tokens need position-based signals to distinguish which image they refer to. Equalizing inter-query distance would destroy multi-image discriminability. Our proof obligation is therefore narrow but exact: removal of the specific failure mode the paper targets, not every possible positional effect. Removing both biases simultaneously while retaining multi-image discriminability is a richer problem we leave to future work.
>
> > **W3 / Q1 / L2-L3** — Pretraining evidence
>
> We fully agree — and appreciate the reviewer's directness — that full pretraining is the decisive experiment. We cannot yet run it, but the current evidence suggests the bias is not simply learned away:
>
> |step|1k|3k|5k|7k|end|
> |-|-|-|-|-|-|
> |Baseline|28.96|27.57|27.80|28.09|27.57|
> |Circle|28.73|27.86|28.44|29.02|28.55|
> |Delta|-0.23|+0.29|+0.64|+0.92|+0.98|
>
> The gap first reflects adaptation cost, then grows monotonically. TAM Func-IoU also improves by +3.45 over the fully pretrained Qwen2.5-VL baseline. Our interpretation: training can learn around a deterministic geometric artifact but cannot remove it, because RoPE rotations contain no learnable parameters — the bias is structural — Huang et al. (2510.23095) independently confirm. The SFT transition lasts ~8.5k steps. We frame the paper as strong SFT evidence plus a clear pretraining hypothesis, not a final answer about full pretraining.
>
> > **Q3** — Failure modes
>
> The weaker regime is OCR-like chart reading, where exact patch-level discriminability matters more than debiasing — these tasks require distinguishing individual patch positions, precisely the radial information CIP sacrifices. **A detailed sub-task breakdown is in our reply to Reviewer yjbo (W3/Q4)**. `dff_rate` is a leftover debugging parameter not used in any experiment; it will be removed from the released code.

---

> > ### Author Rebuttal · Reviewer_jjCq · 2026-04-03
> >
> > Thank you for the thoughtful rebuttal. The additional quantitative analysis and the more precise discussion of PTD, geometry preservation, and failure modes addressed my main concerns. I appreciate the authors’ effort to narrow the claims where needed and to clarify the practical details. Overall, the rebuttal makes the paper stronger, and I maintain my positive evaluation.

---

> > > ### Author Response · Authors · 2026-04-04
> > >
> > > We sincerely thank the reviewer for the detailed and insightful feedback, which has substantially sharpened our presentation. We are grateful that the additional analysis addressed the main concerns, and we appreciate the continued positive evaluation of our work.

---

### Official Review · Reviewer_EzGo · 2026-03-13

**Soundness:** 3
**Presentation:** 3
**Significance:** 3
**Originality:** 4
**Overall Recommendation:** 4
**Confidence:** 3

**Summary:**

This paper introduces Circle-RoPE, a novel positional encoding strategy designed to mitigate cross-modal positional bias in vision–language models. The authors observe that conventional RoPE embeds text and image tokens within a shared positional space, which can lead to unintended attention patterns where token interactions are influenced more by indexing distance than by semantic relevance. To analyze this issue, the paper proposes the Per-Token Distance (PTD) metric to quantify cross-modal positional entanglement and shows that achieving a PTD value close to zero can remove geometric bias in attention. To realize this property, Circle-RoPE remaps 2D image coordinates into a 3D annular structure orthogonal to the text-position axis, forming a cone-like geometric configuration in which each text token maintains equal distance to all image tokens while preserving spatial relationships within the image itself.

The framework is further extended with Alternating Geometry Encoding (AGE), which alternates Circle-RoPE and standard grid-based RoPE across transformer layers to balance unbiased cross-modal interactions with accurate intra-image spatial modeling. The proposed method can be incorporated into existing transformer architectures with minimal modification, making it a drop-in replacement for standard positional embeddings. Experiments across multiple vision–language backbones, including Qwen2.5-VL and LLaVA, demonstrate consistent improvements on spatial grounding and visual reasoning benchmarks. Overall, the paper highlights a previously underexplored limitation of traditional positional encodings in multimodal transformers and proposes a geometrically motivated solution that improves cross-modal attention while maintaining compatibility with existing models.

**Compliance With Llm Reviewing Policy:**

Affirmed.

**Final Justification:**

Thank you to the authors for the thoughtful rebuttal and for clarifying several aspects of the work. I appreciate the effort to address the concerns raised. While some points have been clarified, a few of my original concerns remain partially resolved. Given that my initial score was already relatively favorable and positive, I will maintain my current evaluation.

**Key Questions For Authors:**

My primary question for the authors concerns the general applicability and scalability of Circle-RoPE across different multimodal architectures and task settings. While the paper demonstrates improvements on spatial grounding and reasoning tasks using models such as LLaVA, it would be helpful to understand whether the same positional geometry remains beneficial for other multimodal tasks such as image–text retrieval, long-context multimodal dialogue, or video understanding. In addition, since the approach relies on a specific geometric arrangement to achieve near-zero cross-modal positional bias compared with RoPE, could the authors clarify how sensitive the method is to the choice of hyperparameters, such as the mixing angle and radius, or to changes in embedding dimensionality and image resolution? It would also be interesting to know whether the identified “adaptation cost” could be reduced if Circle-RoPE were incorporated during full pretraining rather than only during supervised fine-tuning, and whether there exists a theoretical framework for scaling these parameters to higher-resolution inputs or varying aspect ratios. Furthermore, regarding the Alternating Geometry Encoding strategy, was there a specific reason for adopting a layer-wise alternation rather than alternatives such as head-wise integration within the same layer? Finally, since the current formulation mainly considers the two-dimensional image setting, it would be useful to understand the computational or geometric challenges that might arise when extending the cone-like projection to higher-dimensional scenarios such as temporal video data, and whether maintaining the PTD ≈ 0 condition would remain the key factor for performance in those settings.

**Limitations:**

A significant limitation of this work is the adaptation cost, as replacing the positional encoding requires a non-trivial amount of optimization (approximately 8.5k steps) before the model's representation geometry stabilizes and begins to outperform standard baselines. Furthermore, due to computational constraints, the evaluation is restricted to supervised fine-tuning (SFT) rather than full-scale pre-training from scratch, which leaves the method's impact on foundational representation learning somewhat unexplored. While the theoretical framework for PTD = 0 is elegant, the current implementation is strictly focused on 2D images; extending the "cone-like" projection to higher-dimensional spatio-temporal data, such as video, remains a theoretical possibility rather than an empirically validated feature. Additionally, the method's reliance on specific hyper-parameters, such as the radius R and mixing angle 𝛼, suggests that optimal performance might require task-specific tuning, potentially limiting its "plug-and-play" versatility across drastically different resolutions or model architectures.

**Strengths And Weaknesses:**

I think this paper has several clear strengths. First, it identifies a very specific but important issue in multimodal transformers—namely the positional coupling between text and image tokens when using RoPE. The analysis using the Per-Token Distance metric is quite insightful because it provides a concrete way to measure cross-modal positional bias rather than just describing the problem qualitatively. Second, the proposed Circle-RoPE design is conceptually elegant; the idea of placing image tokens on a 3D annulus orthogonal to the text axis to equalize cross-modal distances is both geometrically intuitive and easy to implement. Third, the method is practical since it can be inserted into existing transformer architectures with minimal changes, making it a relatively lightweight modification rather than requiring full model retraining. Finally, the empirical results across different backbones, such as Qwen2.5-VL and LLaVA, demonstrate consistent improvements on spatial grounding and reasoning benchmarks, which suggests that the approach is reasonably robust across models.

At the same time, there are a few aspects that could benefit from further discussion. One concern is that the geometric design introduces additional structural assumptions about how modalities should interact, and it is not entirely clear whether the same positional scheme would remain effective for tasks beyond spatial reasoning or grounding. Another limitation is that most of the evaluation focuses on a small number of benchmarks and model backbones, so it would be useful to see broader validation across larger or more diverse multimodal architectures. Finally, while the Alternating Geometry Encoding strategy is an interesting compromise between Circle-RoPE and standard RoPE, the paper could provide deeper analysis of how this alternation influences attention dynamics across layers and whether different scheduling strategies might further improve performance.

---

> ### Author Rebuttal · Authors · 2026-03-31
>
> We thank the reviewer for this constructive evaluation.
>
> > **W1 / Q1** — Assumption
>
> Circle-RoPE targets a general issue: in RoPE-based VLMs, some patches receive higher attention purely from geometric proximity in shared RoPE space. Our assumption is deliberately conservative — a text token should have no geometry-induced preference among patches of one image — weaker than hard sequential embeddings or full M-RoPE.
>
> |Subset|Delta|
> |-|-|
> |MMMU: Energy/Power, Chemistry, Geography, Marketing, CS, Math|+20.0,+10.0,+10.0,+3.3,+6.7,+10.0|
> |MathVista: logical, word, arithmetic|+8.1,+4.3,+2.8|
> |MMStar: Science&Technology|+2.4|
>
> These are not localization subsets — removing cross-modal bias improves knowledge, logic, and math reasoning, not only grounding.
>
> > **W2 / Q2** — Scale & sensitivity
>
> Under our budget we chose two materially different backbones: Qwen2.5-VL-3B (native M-RoPE) and LLaVA-0.5B (1D-RoPE). Circle-RoPE improves both, confirming the effect is not architecture-specific. PTD is determined by positional indices alone, not model depth or hidden size; the same M-RoPE geometry is shared across Qwen 3B/7B variants, so the bias source is architectural. We plan 7B validation with additional compute.
>
> Table 4 shows all Circle-RoPE configurations outperform baseline (avg 46.85): alpha sweep at $R=10$ yields 47.77–47.98, radius sweep shows even less sensitivity (range 0.37). The default $\alpha=0.5, R=10$ transfers directly to LLaVA-0.5B without retuning (avg 27.36 vs. 1D-RoPE 26.95/M-RoPE 26.93, Table 7).
>
> |MMMU max dim|<=224|225-336|337-448|449-672|673-1024|1025-1200|>1200|
> |-|-|-|-|-|-|-|-|
> |Delta|+2.9|+3.7|+5.8|+1.9|+1.8|0|0|
>
> No resolution bucket degrades. This is consistent with CIP: coordinates are recentered and normalized before angular mapping, so the geometry is resolution- and aspect-ratio-aware.
>
> > **Q3** — Adaptation cost
>
> Switching from pretrained M-RoPE to Circle-RoPE requires ~8.5k SFT steps to stabilize. Under full pretraining this cost would largely vanish since the model never needs to unlearn. The MMMU-Pro learning curve (**detailed in our reply to Reviewer jjCq, W3/Q1**) grows from -0.23 at 1k to +0.98 at the end, confirming the cost is temporary.
>
> Formally, $\text{PTD}=0$ ensures the rotation-induced decay in $s_{t,i}=q_t^T R_{m_i-m_t} k_i$ collapses to a shared scalar; after softmax, patch ranking is governed purely by semantic similarity $q_t^T k_i$.
>
> > **W3 / Q4 / Q5 / L1-L4** — AGE & scope
>
> AGE uses a fixed odd/even schedule, not a tuned depth search. Layer-wise attention evidence is **in our reply to Reviewer yjbo (W1/Q2)**: Circle layers produce more uniform $T \to I$ attention (higher entropy, lower CV), while M-RoPE layers preserve grid-sensitive $I \to I$ locality. This alternation has a direct precedent — DiNAT (2209.15001) alternates local and dilated attention — and AGE extends this idea to positional encoding.
>
> We chose layer-wise over head-wise mixing for two reasons. First, if Circle heads signal "$i_a$ and $i_b$ are equidistant to $t$" while M-RoPE heads signal "$i_a$ is closer" within the same layer, the shared output projection $W_O$ must fuse geometrically contradictory sub-vectors, wasting capacity and propagating inconsistent positional signals to the next layer. Second, shared parameters ($W_O$, LayerNorm) receive opposing gradients from the two head groups — one pushes toward position-invariant alignment, the other toward exploiting grid priors. This is the gradient conflict studied in multi-task optimization (PCGrad, 2001.06782), which slows convergence. Third, layer-wise alternation creates a sequential complementary pipeline — each layer applies one coherent geometry, the residual stream accumulates both priors across depth — whereas head-wise forces every layer to produce a compromise representation that is neither fully grid-consistent nor fully circle-consistent, and this ambiguity compounds across layers. Fourth, Circle-RoPE and M-RoPE rotations arise from incompatible coordinate parameterizations; linearly combining their head outputs in $W_O$ breaks RoPE's rotation group structure, producing position vectors in no coherent coordinate frame. Layer-wise alternation avoids all four issues.
>
> Regarding scope: image-text retrieval uses dual-encoder models (CLIP/BLIP) where image and text never share RoPE index space, so the cross-modal bias we address cannot arise. Long-context dialogue is plausible but unvalidated beyond 4096 tokens due to GPU memory. Video is theoretically natural: CIP generalizes by mapping coordinates $(x,y,t)$ onto a sphere in 4D space orthogonal to the text axis. $\text{PTD}=0$ still holds because all visual tokens lie at radius $R$, giving $\|m_i-m_t\|^2=R^2+p_t^2$ regardless of position. The barrier is Qwen2.5-VL's 3D RoPE must expand to 4D, requiring pretraining beyond our compute budget. The 4D derivation with $\text{PTD}=0$ proof will appear in the revised appendix.

---

> > ### Author Rebuttal · Reviewer_EzGo · 2026-04-03
> >
> > Thank you to the authors for the thoughtful rebuttal and for clarifying several aspects of the work. I appreciate the effort to address the concerns raised. While some points have been clarified, a few of my original concerns remain partially resolved. Given that my initial score was already relatively favorable and positive, I will maintain my current evaluation.

---

> > > ### Author Response · Authors · 2026-04-04
> > >
> > > We sincerely thank the reviewer for the thoughtful and constructive feedback throughout the discussion. We are glad that our clarifications have addressed the main concerns, and we greatly appreciate the positive assessment of our work.

---

### Official Review · Reviewer_jKyy · 2026-03-13

**Soundness:** 2
**Presentation:** 3
**Significance:** 2
**Originality:** 3
**Overall Recommendation:** 4
**Confidence:** 4

**Summary:**

This paper studies positional encoding in VLMs and identifies a structural limitation of standard Rotary Position Embedding (RoPE) when applied to multimodal inputs. The submission outlines a notable area concerning cross-modal positional bias caused by concatenating text and image tokens into a shared positional index space. This research attempts to address a pressing problem by proposing Circle-RoPE, a positional encoding method that geometrically decouples image and text token positions. The method introduces a Per-Token Distance (PTD) metric to quantify cross-modal positional interference and proves that PTD = 0 removes geometry-induced attention bias. Circle-RoPE maps image tokens onto an annular structure orthogonal to the text axis, ensuring equal positional distance between text tokens and image tokens while preserving intra-image spatial relationships. An additional Alternating Geometry Encoding (AGE) scheme alternates Circle-RoPE and grid-based RoPE across layers to balance cross-modal alignment and local spatial structure.

**Compliance With Llm Reviewing Policy:**

Affirmed.

**Final Justification:**

Thank you for the authors’ response. After carefully reviewing the updated rebuttal and considering the comments from other reviewers, I have decided to raise my score to 4.

**Key Questions For Authors:**

Please see the weaknesses.

**Limitations:**

The paper discusses the potential negative societal impact of the work but does not provide its limitations. Providing a discussion of the limitations would help readers better understand the scope and potential constraints of the proposed approach.

**Strengths And Weaknesses:**

### Strengths

1. This work correctly identifies that shared positional indexing across modalities introduces spurious geometric bias in RoPE-based VLMs.

2. The introduction of the Per-Token Distance (PTD) metric is a strong conceptual contribution.

3. The Circle-RoPE constructs a cone-like geometry, which text tokens lie on a vertical axis and image tokens lie on an annulus orthogonal to that axis.

### Weaknesses

1. The introduction of AGE is relatively brief. The paper does not clearly explain how the layers are selected to apply Circle-RoPE versus M-RoPE (just experimental analysis), nor what criteria guide this design choice. Providing a clearer description of the layer selection strategy would help readers better understand and reproduce the method.

2. The performance improvements are relatively modest. While consistent, the gains are small.

3. The efficiency of Circle-RoPE is not discussed. Since the method introduces additional geometric transformations in positional encoding, it would be helpful to report the runtime or computational overhead during training and inference.

4. In image–text interleaved scenarios, although Circle-RoPE enforces PTD = 0, it may also weaken the positional alignment between text tokens and their corresponding image regions. By making all image tokens equidistant to a text token, the positional cues that could help establish text–image correspondence may be diminished.

---

> ### Author Rebuttal · Authors · 2026-03-31
>
> We sincerely thank the reviewer for this careful and constructive review.
>
> > **W1** — AGE layer schedule
>
> The reviewer is right that our AGE description was insufficiently systematic — we have revised Section 4.2 accordingly. AGE uses a fixed odd/even schedule: odd layers apply M-RoPE, even layers apply Circle-RoPE. The guiding principle is to maximize geometric diversity across adjacent layers while keeping each layer internally coherent.
>
> |Strategy|MMMU|MMMU_Pro|MathVista|AI2D|RealWorldQA|Avg|
> |-|-|-|-|-|-|-|
> |All-M-RoPE|51.56|28.01|62.40|79.22|66.10|57.46|
> |All-Circle|51.32|28.42|65.20|80.39|66.34|58.33|
> |Upper-only|52.66|28.51|65.20|79.80|65.94|58.42|
> |Lower-only|53.48|28.62|64.50|79.30|66.32|58.44|
> |AGE|52.11|28.44|63.40|81.80|66.54|58.46|
>
> All Circle-based settings outperform the M-RoPE baseline (+0.87 to +1.00), and the three mixed schedules are nearly tied (58.42/58.44/58.46). This confirms the benefit comes from mixing geometries rather than a specific layer split. Lower-only Circle excels on MMMU/MMMU-Pro (early debiasing), upper-only on MathVista (late-stage reasoning), and AGE achieves the best overall by combining both across depth. Layer-wise attention statistics (**detailed in our reply to Reviewer yjbo, W1/Q2**) confirm Circle layers produce more uniform $T \to I$ attention without degrading $I \to I$ locality, validating AGE as a principled geometric decomposition. We chose layer-wise over head-wise mixing to avoid fusing contradictory positional operators in shared projections — **the detailed theoretical argument is in our reply to Reviewer EzGo (W3/Q4)**.
>
> > **W2** — Magnitude of gains
>
> We appreciate this honest assessment and fully understand the concern. The average gain (+1.00) is numerically moderate, but we believe it is meaningful. First, Circle-RoPE improves every main benchmark — a consistency rare for PE methods. Second, the largest gains appear where the mechanism predicts: AI2D +2.58 (spatial diagram understanding) and TAM Func-IoU +3.45 (functional grounding, Table 6). Third, Circle-RoPE adds zero learned parameters and essentially zero runtime cost (see W3), so the gain is purely from better geometry, mirroring VRoPE (2502.11664) for video. Fourth, Table 5 isolates the structural contribution: Circle-RoPE 57.76 vs. Unordered 54.67, a +3.09 gap even though both achieve $\text{PTD}=0$. Fifth, the signal is stable across random data orderings (4 independent 800-step runs, 95% CIs):
>
> |AI2D|MMStar|MMMU-Pro(vis)|MMMU-Pro(10c)|MMMU(val)|
> |-|-|-|-|-|
> |.7592±.0026|.5560±.0031|.1864±.0034|.2973±.0024|.4263±.0039|
>
> Max CI width is 0.0078 (MMMU-val), confirming reproducibility.
>
> > **W3** — Computational overhead
>
> We thank the reviewer for highlighting this important omission. Circle-RoPE only changes positional indices before the Transformer forward pass; it does not alter attention complexity or add learned modules:
>
> |Setting|Train step ms|tok/s|Prefill ms|Decode ms/tok|
> |-|-|-|-|-|
> |M-RoPE|327.0|4868|38.5|15.5|
> |Circle+AGE|328.1|4853|38.2|15.7|
>
> *Measured on RTX PRO 6000 Blackwell, batch 4, 1×448×448.* Training overhead: +1.1ms/step; inference: prefill -0.3ms, decode +0.2ms/token. Peak memory unchanged. CIP uses closed-form $O(N)$ operations (Eqs. 5–16) that can be cached per resolution.
>
> > **W4** — Multi-image
>
> This is a perceptive concern, and we are grateful for the opportunity to clarify. The worry would be valid if all images shared one ring — but this is not the case. As described in Appendix B.7, each image receives its own ring, with centers translated along $g=[1,1,1]^T$:
>
> $c_i^{\text{final}} = c_i + (i-1) \cdot \Delta \cdot g$ (default $\Delta=1$), e.g., three images at $[0,0,0], [1,1,1], [2,2,2]$
>
> $\text{PTD}=0$ is enforced per-image: for a fixed text token $t$, $d(t,i_a)=d(t,i_b)$ for patches within the same image, but the overall distance to different images varies. This preserves inter-image discriminability while removing intra-image bias. By contrast, M-RoPE places all images on a shared sequential index, so adjacent images' boundary patches receive similar indices — creating spurious cross-image similarity that per-image rings eliminate.
>
> > **L1** — Limitations
>
> We fully agree with the reviewer that stating boundaries explicitly makes the paper stronger. We have added a limitations section covering: (1) SFT-only evidence — full pretraining remains an open hypothesis, though the widening learning curve gap (-0.23 to +0.98) and TAM improvement suggest the bias persists; (2) 2D-image scope — video requires 4D RoPE extension (**see Reviewer EzGo, W3/Q5**); (3) OCR-heavy chart reading is a weaker regime (**see Reviewer yjbo, W3/Q4**: scientific -2.46, statistical -1.99, ChartQA -0.92); (4) adaptation cost of ~8.5k SFT steps when switching from M-RoPE (eliminated under pretraining); (5) very small objects (<4 patches) may have insufficient angular resolution on the ring.

---

> > ### Author Rebuttal · Reviewer_jKyy · 2026-04-03
> >
> > Thank you for the authors’ response and for the detailed explanation of AGE, as well as addressing my concerns regarding multi-image scenarios. However, I find the improvements brought by the method to be limited. Overall, I maintain my original rating.

---

> > > ### Author Response · Authors · 2026-04-03
> > >
> > > We sincerely thank the reviewer for carefully reading our response and for acknowledging the clarifications on AGE and multi-image scenarios. We fully appreciate the concern regarding the magnitude of improvements and would like to offer a further perspective grounded in the nature of our method and the conditions under which our results were obtained.
> > >
> > > Circle-RoPE is, at its core, a structural correction to a foundational VLM component. RoPE serves as core infrastructure in mainstream VLM architectures, and the cross-modal coupling bias we address is a *deterministic geometric defect* — RoPE rotations contain no learnable parameters, making this bias structural and impossible to "learn away" through training (independently corroborated by Huang et al., 2510.23095). Improvements at this foundation layer release the model's overall potential rather than optimizing for any particular benchmark, and we believe that for such methods, **cross-task consistency, mechanistic interpretability, and efficiency** are more informative criteria than the absolute magnitude of any single metric.
> > >
> > > It is particularly worth noting that our experiments were conducted under **highly constrained resources**: only one-tenth of MAmmoTH-VL Instruct (1M samples, excluding video), SFT only rather than full pretraining. Despite this, Circle-RoPE achieves consistent gains across all five major benchmarks (*average +1.00*), introduces zero learnable parameters, and adds only +1.1ms/step training overhead with negligible inference cost. This fact is itself highly informative: if cross-modal positional bias were a problem that sufficient training could dissolve, then given that Qwen2.5-VL already underwent large-scale pretraining with M-RoPE, minimal SFT data should not yield sustained additional gains. The MMMU-Pro learning curve (**detailed in our reply to Reviewer jjCq, W3/Q1**) reinforces this — Circle-RoPE's advantage grows monotonically from −0.23 at 1k steps to +0.98 at the end of training, with no saturation, suggesting the method's potential would be more fully realized under larger-scale training.
> > >
> > > From a mechanistic standpoint, the TAM result is especially telling: **+3.45 in Func-IoU** directly measures whether text tokens can locate semantically relevant image tokens without geometric bias — precisely what Circle-RoPE addresses. In VLMs, accurate text-to-image grounding underpins visual understanding, spatial reasoning, and knowledge-based QA alike, so this substantial mechanistic improvement validates that our method enhances the model's most fundamental cross-modal alignment capability. Importantly, benefits are not confined to spatial tasks. **As shown in our reply to Reviewer EzGo (W1/Q1)**, Circle-RoPE yields notable gains on knowledge- and reasoning-oriented subtasks: MMMU Energy/Power (+20.0), Chemistry (+10.0), Math (+10.0), MathVista logical reasoning (+8.1), and word problems (+4.3), demonstrating that removing cross-modal positional bias produces broad, category-spanning improvements.
> > >
> > > The structural ablation in Table 5 further underscores Circle-RoPE's design: Unordered embedding, which also achieves $PTD = 0$, causes severe degradation (Avg 54.67 vs. Circle-RoPE 57.76, gap of *3.09*), proving that gains do not arise from simply eliminating distance variance but critically depend on preserving intra-image spatial structure while removing cross-modal bias. Circle-RoPE also achieves the best performance on the architecturally distinct LLaVA-0.5B (Avg 27.36 vs. 1D-RoPE 26.95 / M-RoPE 26.93, Table 7) using the same hyperparameters from Qwen2.5-VL without architecture-specific tuning — further evidence that gains originate from a fundamental positional encoding improvement rather than model-specific overfitting.
> > >
> > > In summary, we fully respect the reviewer's attention to numerical margins. However, taken together — consistent gains across all benchmarks under severely constrained resources, +3.45 in the core grounding metric Func-IoU, broad improvements spanning knowledge / reasoning / spatial task families, a pure geometric correction with zero parameters and zero overhead, a monotonically widening learning curve pointing to greater potential at scale, and architecture-agnostic generalization — we believe these collectively demonstrate that Circle-RoPE constitutes a **substantive and efficient improvement** to VLM positional encoding infrastructure, whose value lies in elevating the model's foundational capacity at minimal cost. We respectfully ask the reviewer to reconsider the score in light of this combined evidence.

---

### Decision · Program_Chairs · 2026-04-30

**Decision:**

Accept (regular)

**Comment:**

This paper received a mixed but overall positive set of reviews. Reviewers generally agreed that it identifies a meaningful and underexplored issue in RoPE-based vision–language models, and that the proposed PTD metric and Circle-RoPE design provide a clear and practically useful way to reason about cross-modal positional bias. The method was also viewed as lightweight and easy to integrate into existing VLMs, with consistent empirical improvements across multiple backbones and benchmarks.

The main concerns were about the relatively modest size of the gains, the need for clearer discussion of structure preservation and failure modes, and whether the current evidence is sufficient to fully justify the broader geometric claims. The rebuttal was helpful in this regard: it clarified the AGE schedule, added efficiency measurements, sharpened the claims around what structure is and is not preserved, and provided more concrete discussion of limitations and weaker regimes. One reviewer explicitly raised their score after the rebuttal, two positive reviewers maintained their favorable assessments, and the remaining weaker review stayed cautious but did not overturn the overall picture.